# Neural Injective Functions for Multisets, Measures and Graphs via a Finite Witness Theorem

**Tal Amir**[1]  **Steven J. Gortler**[2]  **Ilai Avni**[1]  **Ravina Ravina**[1]  **Nadav Dym**[1,3]

[1] Faculty of Mathematics, Technion – Israel Institute of Technology, Haifa, Israel
[2] School of Engineering and Applied Sciences, Harvard University, Cambridge, USA
[3] Faculty of Computer Science, Technion – Israel Institute of Technology, Haifa, Israel.

## Abstract

Injective multiset functions have a key role in the theoretical study of machine learning on multisets and graphs. Yet, there remains a gap between the provably injective multiset functions considered in theory, which typically rely on polynomial moments, and the multiset functions used in practice, which rely on *neural moments* — whose injectivity on multisets has not been studied to date.

In this paper, we bridge this gap by showing that moments of neural networks do define injective multiset functions, provided that an analytic non-polynomial activation is used. The number of moments required by our theory is optimal essentially up to a multiplicative factor of two. To prove this result, we state and prove a *finite witness theorem*, which is of independent interest.

As a corollary to our main theorem, we derive new approximation results for functions on multisets and measures, and new separation results for graph neural networks. We also provide two negative results: (1) moments of piecewise-linear neural networks cannot be injective multiset functions; and (2) even when moment-based multiset functions are injective, they can never be bi-Lipschitz.

## 1  Introduction

Multisets are a slight generalization of sets: like sets, they are an unordered collection of elements $\{\!\{x_1, \ldots, x_k\}\!\}$, but unlike sets, repetitions are allowed. Multisets arise naturally in many machine-learning tasks. They are the natural way to represent point clouds in $\mathbb{R}^3$, neighborhoods of vertices in graphs, and any other data that has an intrinsic order that is immaterial to the task at hand.

We refer to functions and architectures whose inputs are multisets in $\mathbb{R}^d$ as *multiset functions* and *multiset architectures*. By definition, these functions do not depend on the order in which the multiset elements are given. This is important not only because the order is irrelevant and thus should not affect the output, but also because otherwise a model may overfit the training data by making predictions based on its intrinsic order.

Multiset architectures are typically constructed using a combination of permutation-invariant operations such as sum- and max-pooling [34], attention mechanisms [23] and sorting [46]. One simple and popular approach, pioneered in the seminal Deep-Sets paper [45], employs multiset functions based solely on sum-pooling. Namely, if the elements of all multisets come from some fixed *alphabet* $\Omega$, any function $f : \Omega \to \mathbb{R}^m$ induces a multiset function $\hat{f}$, to which we refer as the *moment* of $f$:

$$\hat{f}\left(\{\!\{x_1, \ldots, x_k\}\!\}\right) = \sum_{i=1}^{k} f(x_i). \tag{1}$$

37th Conference on Neural Information Processing Systems (NeurIPS 2023).

While moment functions of the form (1) are simple, they are quite powerful. For example, in [45] it was shown that if $\Omega$ is countable, and the multisets have no repetitions (so they are just sets), then for an appropriate $f : \Omega \to \mathbb{R}$, the induced function $\hat{f}$ maps the input sets *injectively* to $\mathbb{R}$.

Injectivity is indeed a desired property for multiset functions. The search for such functions stems from the quest to find an architecture that can approximate *all* multiset functions. Clearly, if $\hat{f}$ assigns the same value $\hat{f}(S_1) = \hat{f}(S_2)$ to two different multisets $S_1 \neq S_2$, then any architecture based on $\hat{f}$ will yield a poor approximation of a multiset function that assigns different values to $S_1$ and $S_2$.

The authors of [45] showed that injectivity is not only necessary for approximation, but also sufficient: Under the assumption that the alphabet $\Omega$ is countable, if the moment $\hat{f}$ of $f$ maps multisets injectively to $\mathbb{R}^m$, then any multiset function $F$ can be written as a composition of the form $F(\{\!\{\boldsymbol{x}_1, \ldots, \boldsymbol{x}_k\}\!\}) = g\left(\sum_{i=1}^{k} f(\boldsymbol{x}_i)\right)$. Motivated by this observation, the authors proposed a neural architecture of this form, with the functions $f$ and $g$ replaced by Multi-Layer Perceptrons (MLPs). This step was justified by the universal approximation power of MLPs.

These intriguing results inspired further research, mainly focusing on seeking injective multiset functions of the form (1) for *continuous* alphabets such as $\Omega = \mathbb{R}^d$. Preferably, such functions should (a) have a minimal *embedding dimension* $m$ while ensuring that $f : \Omega \to \mathbb{R}^m$ induces an injective $\hat{f}$; and (b) be practical to compute. We next summarize some of these results:

For a countable $\Omega$, the Deep-Sets paper as well as [44] showed that an embedding dimension $m = 1$ is sufficient. For $\Omega = \mathbb{R}$, if the multisets contain at most $n$ elements, and $f$ is continuous, then an embedding dimension of $m \geq n$ is necessary and sufficient for injectivity [41, 6, 42].

For $\Omega = \mathbb{R}^d$ and multisets of size at most $n$, it was shown in [16] that $m \geq nd$ is necessary for injectivity. As for an upper bound on the required $m$, while some polynomials discussed in the literature achieve injectivity with a rather high exponential [3, 26, 37] or polynomial [43] dimension $m$, recent work [9] achieved injectivity with $m = 2nd + 1$, using a polynomial $f$ with randomly chosen coefficients — thus achieving the lower bound essentially up to a multiplicative factor of $2$.

While the above works provide injective multiset functions with optimal or near-optimal embedding dimension, these functions are typically polynomials, and not the MLPs used in practice. As mentioned above, many papers [45, 44, 26] justify this by the fact that MLPs can approximate any function, and thus any polynomial. However, using this argument, we have no control on the number of neurons required for injectivity — which in some cases may be infinite, as we show in Section 4. In this paper, we address this limitation by providing a practical and efficient method to construct functions of the form (1) that are provably injective while having a near-optimal number of neurons. We now state this formally.

## 1.1 Problem Statement

Let $\Omega \subseteq \mathbb{R}^d$ be a set, to which we refer as an *alphabet*. Denote by $\mathcal{S}_{\leq n}(\Omega)$ the collection of all multisets $\{\!\{\boldsymbol{x}_1, \ldots, \boldsymbol{x}_k\}\!\}$ with $\boldsymbol{x}_1, \ldots, \boldsymbol{x}_k \in \Omega$ and $k \leq n$. Any function $f : \Omega \to \mathbb{R}^m$ induces a *moment function* $\hat{f} : \mathcal{S}_{\leq n}(\Omega) \to \mathbb{R}^m$ as in (1). If $\hat{f}$ is injective, we say that $f$ is *moment injective* on $\mathcal{S}_{\leq n}(\Omega)$.

We also consider a natural generalization from multisets to measures, by identifying each multiset $\{\!\{\boldsymbol{x}_1, \ldots, \boldsymbol{x}_k\}\!\}$ with the measure $\mu = \sum_{i=1}^{k} \delta_{\boldsymbol{x}_i}$, where $\delta_{\boldsymbol{x}}$ is the Dirac measure that assigns a unit weight to $\boldsymbol{x}$. In this generalized setting, the induced multiset function $\hat{f}$ of (1) is just the integral of $f$ with respect to the measure $\mu$. More generally, we consider signed measures $\mu = \sum_{i=1}^{n} w_i \delta_{\boldsymbol{x}_i}$, with weights $w_i \in \mathbb{R}$ that can be negative, and points $\boldsymbol{x}_i$ that belong to an alphabet $\Omega \subseteq \mathbb{R}^d$. We denote the space[1] of all such measures by $\mathcal{M}_{\leq n}(\Omega)$. A function $f : \Omega \to \mathbb{R}^m$ induces a moment function $\hat{f} : \mathcal{M}_{\leq n}(\Omega) \to \mathbb{R}^m$ defined by

$$\hat{f}(\mu) = \int_\Omega f(\boldsymbol{x})d\mu(\boldsymbol{x}) = \sum_{i=1}^{n} w_i f(\boldsymbol{x}_i), \quad \text{where} \quad \mu = \sum_{i=1}^{n} w_i \delta_{\boldsymbol{x}_i}. \tag{2}$$

---

[1]While we use the term *space* for $\mathcal{S}_{\leq n}(\Omega)$ and $\mathcal{M}_{\leq n}(\Omega)$, note that these are not vector spaces, since the sum of two measures in these spaces might be supported on more than $n$ points.

| Domain | $\mathcal{M}_{\leq n}(\mathbb{R}^d)$ | $\mathcal{S}_{\leq n}(\mathbb{R}^d)$ | $\mathcal{M}_{\leq n}(\Sigma)$ | $\mathcal{S}_{\leq n}(\mathbb{Z}^d)$ | $\mathcal{S}_{\leq n}(\Sigma_\alpha)$ |
|---|---|---|---|---|---|
| Analytic activation | $2n(d+1)+1$ | $2nd+1$ | $2n+1$ | $1$ | $1$ |
| Piecewise-linear activation | $\infty$ | $\infty$ | $\infty$ | $\infty$ | $1$ |
| Lower bound | $n(d+1)$ | $nd$ | $n$ | $1$ | $1$ |

**Table 1:** The embedding dimension required for constructing injective functions of measures and multisets. $\Sigma \subset \mathbb{R}^d$ is any infinite countable alphabet. First row: dimensions for which our theorems guarantee injectivity when using analytic non-polynomial activations. Second row: with infinite alphabets, moments of a neural network of any finite size with a piecewise-linear activation cannot be injective, except in the multiset case, with some special countable alphabets such as $\Sigma_\alpha$, defined in Appendix B.1. Third row: lower bounds on the embedding dimension required for injectivity. These bounds show that our results from the first row of the table are optimal essentially up to a factor of two.

If $\hat{f}$ is injective, we say that $f$ is moment-injective on $\mathcal{M}_{\leq n}(\Omega)$. Naturally, injectivity on $\mathcal{M}_{\leq n}(\Omega)$ implies injectivity on subsets of this space, such as the space of measures in $\mathcal{M}_{\leq n}(\Omega)$ that are probability measures, or that have only positive weights. In particular, if $f$ is moment-injective on $\mathcal{M}_{\leq n}(\Omega)$, then it is moment-injective on $\mathcal{S}_{\leq n}(\Omega)$.

To summarize, the main questions we focus on in this paper are:

**Main Questions:** (a) Under what conditions is an MLP $f$ moment-injective on spaces of multisets $\mathcal{S}_{\leq n}(\Omega)$ or measures $\mathcal{M}_{\leq n}(\Omega)$? (b) How many neurons are needed to achieve this injectivity?

## 2 Main Results

Interestingly, we find that the answers to these two questions largely depend on the activation function. Consider shallow neural networks $f : \mathbb{R}^d \to \mathbb{R}^m$ of the form

$$f(\boldsymbol{x}; \boldsymbol{A}, \boldsymbol{b}) = \sigma(\boldsymbol{A}\boldsymbol{x} + \boldsymbol{b}), \quad \boldsymbol{A} \in \mathbb{R}^{m \times d}, \ \boldsymbol{b} \in \mathbb{R}^m, \tag{3}$$

with the activation function $\sigma : \mathbb{R} \to \mathbb{R}$ applied entrywise to $\boldsymbol{A}\boldsymbol{x} + \boldsymbol{b}$. Suppose that $\sigma$ is analytic and non-polynomial; such activations include the sigmoid, softplus, tanh, swish and sin. In Section 3 we show that for a large enough $m$, such networks $f(\boldsymbol{x}; \boldsymbol{A}, \boldsymbol{b})$ with random parameters $\boldsymbol{A}, \boldsymbol{b}$ are moment-injective on $\mathcal{M}_{\leq n}(\Omega)$ and on $\mathcal{S}_{\leq n}(\Omega)$; namely, their induced moment functions $\hat{f}$ of (2) are injective. This holds for various natural choices of $\Omega$.

The embedding dimension $m$ required in (3) depends on the dimension $d$ of $\Omega$: For $\Omega = \mathbb{R}^d$, to achieve injectivity on $\mathcal{S}_{\leq n}(\Omega)$ or $\mathcal{M}_{\leq n}(\Omega)$, it suffices to take $m = 2nd + 1$ or $m = 2n(d+1)+1$ respectively. When $\Omega$ is countable, $m = 1$ or $m = 2n + 1$ are sufficient (corresponding to $d = 0$). In Appendix C we show that in all these cases, these embedding dimensions are optimal essentially up to a multiplicative factor of two. These results are summarized in Table 1. In Appendix C we also discuss examples where the optimal embedding cardinality for $\mathcal{M}_{\leq n}(\mathbb{R})$ is obtained.

At the core of our poof of moment injectivity is a theorem which we name the *finite witness theorem*. This theorem enables reducing an infinite family of analytic equality constraints $\{F(\boldsymbol{x}; \boldsymbol{\theta}) = 0 \mid \boldsymbol{\theta} \in \mathbb{W}\}$ to a finite subset $\{F(\boldsymbol{x}; \boldsymbol{\theta}_i) = 0 \mid i = 1, \ldots, m\}$. This theorem generalizes the results in [9], where a special case of it was proved for semialgebraic domains and functions. The theorem we prove here (see Appendix A) applies to a much wider class of domains and functions, among which are analytic functions. In addition to our main result, we use the finite witness theorem to prove moment injectivity of Gaussian functions (Proposition 3.5) and deep MLPs (Proposition 3.6), and we believe it shall find additional applications beyond those discussed in this work.

**Negative Results** We also prove two negative results for moment-based multiset functions: We show that in contrast to analytic activations, with piecewise linear (PwL) activations, such as ReLU, leaky ReLU and hard arctan, moment injectivity on spaces of measures $\mathcal{M}_{\leq n}(\Omega)$ with infinite $\Omega$ is *impossible*. On multiset spaces $\mathcal{S}_{\leq n}(\Omega)$, moment injectivity with PwL activations can be obtained for some irregular, countable $\Omega$, such as the alphabet $\Omega = \Sigma_\alpha$ defined in Appendix B, but not for infinite alphabets that arise naturally, like $\Omega = \mathbb{R}^d$, $\mathbb{Z}^d$ or $(0, 1)^d$. These results are summarized in the bottom row of Table 1. The second negative result is that while moments of MLPs with analytic activations can be injective, they can never be stable in the bi-Lipschitz sense. This points to a possible advantage

| | Analytic | | | PwL | | |
|---|---|---|---|---|---|---|
| Dim | Tanh | SiLU | Sin | HardTanh | ReLU | Leaky |
| 1 | 0 | 0 | 0 | 7 | 17 | 7 |
| 10 | 0 | 0 | 0 | 3 | 7 | 7 |
| 50 | 0 | 0 | 0 | 4 | 5 | 5 |
| 100 | 0 | 0 | 0 | 1 | 0 | 0 |

(a)

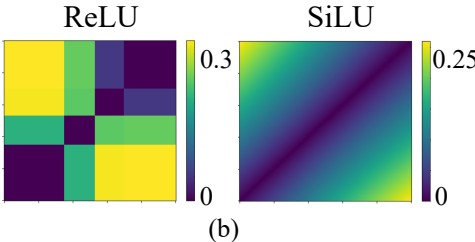

(b)

**Figure 1:** (a) The number of failures of graph neural networks, with varying hidden dimension and activation, to achieve WL separation on the 600 graphs from the TUDataset [30]. Analytic activations succeed on all graphs, as Theorem 6.3 predicts. (b) The normalized smallest singular value of multiset functions induced by piecewise-linear ReLU-networks and analytic SiLU-networks. Piecewise-linear networks have singularities on squares intersecting the diagonal, leading to non-injectivity. Analytic networks are moment injective, but have singularities on the diagonal, which leads to a non-Lipschitz inverse. See the end of Section 5 for more details.

of injective multiset functions that are not based on moments, but rather on sorting [3] or max-filters [5]. These multiset functions are not only injective but also bi-Lipschitz.

**Implications for learning on multisets and graphs** The result on moment injectivity of MLPs with analytic non-polynomial activation enables us to improve upon two seminal theoretical results in the study of functions on multisets and graphs:

**(a) Universality for multisets.** In Corollaries 6.1 and 6.2, we show that any continuous function on a space of multisets or measures respectively can be presented as a continuous vector-to-vector function composed with a moment function $\hat{f}$ of an MLP of the form (3). The MLP has the same embedding dimension $m$ as in Table 1. Essentially, this result replaces the moment-injective polynomials traditionally used in the characterization of multiset functions [45, 42] by MLPs.

**(b) Separation power of Graph Neural Networks.** Famously, the ability of *Message-Passing Neural Networks (MPNNs)* to separate distinct graphs is at most that of the Weisfeiler-Leman (WL) graph isomorphism test, with equivalence taking place if the multiset functions used in the MPNN are injective [44]. Injective multiset functions are also used in generalizations of this result, such as the equivalence of high-order *Graph Neural Networks (GNNs)* to high-order WL tests [31, 26], and recent results on geometric GNNs and their corresponding WL tests [15, 16, 25, 33, 8].

Using the fact that an embedding dimension of one is sufficient to achieve injectivity on $\mathcal{S}_{\leq n}(\Omega)$ with countable $\Omega$, we show in Theorem 6.3 that standard MPNNs with analytic non-polynomial activations and random parameters have the separation power of WL, even when their architecture only uses a single feature per node. This can be compared on the one hand with the construction in [44], which also requires a single node feature but uses multiset aggregators that are not MLPs, and on the other hand with works that do consider MLPs with ReLU activations [31, 1], but require a number of node features and parameters that depends polylogarithmically on the number of nodes. In contrast, our construction requires a single node feature and a fixed number of parameters (though we have no bound on the number of bits required for achieving separation using floating-point arithmetic). A numerical verification of these results is shown in Figure 1(a), where we show that, on the 600 graphs in the TUDataset [30], MPNNs with three different analytic activations were equivalent to the WL-test even with a single node feature, whereas three different PwL activations were in some cases weaker than WL when a small number of node features was used.

Independently of this work, it was recently proved in [17] that MPNNs with certain analytic activations can separate all trees of depth two, while separation fails with PwL and even piecewise-polynomial activations. Our results here are stronger in that we show separation for *all* graphs separable by WL, and *all* analytic non-polynomial activations.

## 2.1 Notation

We denote vectors by boldface letters, e.g. $\boldsymbol{x}, \boldsymbol{y}$, and scalars by plain letters $x, y$. The inner product of $\boldsymbol{a}, \boldsymbol{x}$ is denoted by $\boldsymbol{a} \cdot \boldsymbol{x}$. Throughout this work, the term *measure* always refers to signed measures.

# 3 Moment injectivity with analytic activations

In this section, we prove moment injectivity for MLPs with analytic non-polynomial activations. We begin by showing that for any non-polynomial function $\sigma : \mathbb{R} \to \mathbb{R}$, a measure $\mu \in \mathcal{M}_{\leq n}(\Omega)$ is uniquely determined by the integrals of all functions $\{\sigma(\boldsymbol{a} \cdot \boldsymbol{x} + b) \mid \boldsymbol{a} \in \mathbb{R}^d, b \in \mathbb{R}\}$. When this holds, we say that $\sigma$ is *discriminatory*:

**Definition 3.1.** Let $\sigma : \mathbb{R} \to \mathbb{R}$ be a continuous function. We say that $\sigma$ is *discriminatory* if for any two signed Borel measures $\mu, \mu'$ on $\mathbb{R}^d$ that are distinct (i.e. $\mu \neq \mu'$), finite (i.e. $|\mu(A)|, |\mu'(A)| < \infty$ for all Borel $A \subseteq \mathbb{R}^d$) and compactly supported, there exist $\boldsymbol{a} \in \mathbb{R}^d$, $b \in \mathbb{R}$ such that

$$\int_{\mathbb{R}^d} \sigma(\boldsymbol{a} \cdot \boldsymbol{x} + b) d\mu(\boldsymbol{x}) \neq \int_{\mathbb{R}^d} \sigma(\boldsymbol{a} \cdot \boldsymbol{x} + b) d\mu'(\boldsymbol{x}). \tag{4}$$

The definition of discriminatory activation functions comes from[2] Cybenko's celebrated paper on the universality of MLPs [7], where it was proved that sigmoid-like activations are discriminatory. This, in turn, was used to prove the universality of MLPs with such activations. In later papers [24, 32] it was shown that universality can be achieved by *all* continuous non-polynomial activations. In the following simple proposition, we use a reverse argument to that used by Cybenko, and show that activations that allow for universality are automatically discriminatory:

**Proposition 3.2.** *Let $\sigma : \mathbb{R} \to \mathbb{R}$ be a continuous function that is not a polynomial; then $\sigma$ is discriminatory.*

*Proof idea.* Suppose that $\int \sigma(\boldsymbol{a} \cdot \boldsymbol{x} + b) d\mu = \int \sigma(\boldsymbol{a} \cdot \boldsymbol{x} + b) d\mu'$ for all $\boldsymbol{a}, b$. By the universality theorem for shallow MLPs [32], all continuous functions can be approximated by linear combinations of functions of the form $\sigma(\boldsymbol{a} \cdot \boldsymbol{x} + b)$. Thus, for any continuous function $f$, $\int f d\mu = \int f d\mu'$. Since a measure is uniquely determined by its integrals of all continuous functions, $\mu$ is equal to $\mu'$. $\square$

Next, we shall prove our main result: If $\sigma$ is *analytic* and discriminatory, then shallow MLPs of reasonable width with $\sigma$ as activation are moment injective.

**Theorem 3.3.** *Let $\sigma : \mathbb{R} \to \mathbb{R}$ be an analytic non-polynomial function. Let $n, d \in \mathbb{N}$, and set $m = 2n(d + 1) + 1$. Then for Lebesgue almost any $\boldsymbol{A} \in \mathbb{R}^{m \times d}, \boldsymbol{b} \in \mathbb{R}^m$, the shallow MLP $f(\boldsymbol{x}) = \sigma(\boldsymbol{A} \cdot \boldsymbol{x} + \boldsymbol{b})$ is moment injective on $\mathcal{M}_{\leq n}(\mathbb{R}^d)$; namely, the function $\hat{f} : \mathcal{M}_{\leq n}(\Omega) \to \mathbb{R}^m$ given by*

$$\hat{f}(\mu) = \sum_{i=1}^{n} w_i \sigma(\boldsymbol{A} \boldsymbol{x}_i + \boldsymbol{b}) \ \text{ for } \ \mu = \sum_{i=1}^{n} w_i \delta_{\boldsymbol{x}_i} \tag{5}$$

*is injective.*

*For moment injectivity on $\mathcal{S}_{\leq n}(\mathbb{R}^d)$, it suffices to take $m = 2nd + 1$. For $\mathcal{M}_{\leq n}(\Sigma)$ or $\mathcal{S}_{\leq n}(\Sigma)$ with countable $\Sigma$, $m = 2n + 1$ and $m = 1$ respectively are sufficient.*

Our proof of Theorem 3.3 is based on a separate theorem, which we name the *finite witness theorem*. This theorem enables us to show that, since any two measures can be discriminated by an integral $\int \sigma(\boldsymbol{a} \cdot \boldsymbol{x} + b) d\mu(\boldsymbol{x})$ for some choice of parameters $\boldsymbol{a}, b$, there exists a finite number of *witness* parameters $(\boldsymbol{a}_i, b_i)_{i=1}^{m}$ that are sufficient for discriminating between *any* two measures. This holds under the assumption that the number of points in both measures is bounded. We shall now state a simple version of this theorem, which suffices for proving Theorem 3.3.

**Theorem 3.4.** *(Finite Witness Theorem, simple version) Let $\mathbb{M} \subseteq \mathbb{R}^L$ be a countable union of affine sets, each of which is of dimension $\leq D$. Let $\mathbb{W} \subseteq \mathbb{R}^{D_\theta}$ be open and connected. Let $F(\boldsymbol{x}; \boldsymbol{\theta}) : \mathbb{M} \times \mathbb{W} \to \mathbb{R}$ be an analytic function. Then for almost any $(\boldsymbol{\theta}^{(1)}, \dots, \boldsymbol{\theta}^{(2D+1)}) \in \mathbb{W}^{2D+1}$, the following set equality holds:*

$$\{(\boldsymbol{x}, \boldsymbol{y}) \in \mathbb{M} \times \mathbb{M} \mid F(\boldsymbol{x}; \boldsymbol{\theta}) = F(\boldsymbol{y}; \boldsymbol{\theta}), \forall \boldsymbol{\theta} \in \mathbb{W}\} =$$
$$\{(\boldsymbol{x}, \boldsymbol{y}) \in \mathbb{M} \times \mathbb{M} \mid F(\boldsymbol{x}; \boldsymbol{\theta}^{(i)}) = F(\boldsymbol{y}; \boldsymbol{\theta}^{(i)}), \forall i = 1, \dots 2D + 1\}.$$

---

[2] with a minor change: we do not require that all measures considered are supported on the same fixed compact set.

Using the finite witness theorem, we are now ready to prove Theorem 3.3.

*Proof of Theorem 3.3.* Recall that a signed measure $\mu \in \mathcal{M}_{\leq n}(\mathbb{R}^d)$ can be parameterized, albeit not uniquely, by a matrix $\boldsymbol{X} = (\boldsymbol{x}_1, \ldots, \boldsymbol{x}_n) \in \mathbb{R}^{d \times n}$ representing $n$ points in $\mathbb{R}^d$, and a weight vector $\boldsymbol{w} = (w_1, \ldots, w_n)$, such that $\mu = \sum_{i=1}^{n} w_i \delta_{\boldsymbol{x}_i}$. Let $\mathbb{M}$ be the space of measure parameters

$$\mathbb{M} = \{(\boldsymbol{w}, \boldsymbol{X}) \in \mathbb{R}^n \times \mathbb{R}^{d \times n}\}.$$

Similarly, let $\mathbb{W}$ be the space of parameters

$$\mathbb{W} = \{(\boldsymbol{a}, b) \in \mathbb{R}^d \times \mathbb{R}\}.$$

Define $F : \mathbb{M} \times \mathbb{W} \to \mathbb{R}$ by

$$F(\boldsymbol{w}, \boldsymbol{X}; \boldsymbol{a}, b) = \sum_{i=1}^{n} w_i \sigma(\boldsymbol{a} \cdot \boldsymbol{x}_i + b). \tag{6}$$

We now invoke the finite witness theorem. Set $m = 2n(d+1)+1$, and note that $m = 2\dim(\mathbb{M})+1$. Recall that $F$ is analytic. According to Theorem 3.4, for almost any choice of $(\boldsymbol{a}_i, b_i)_{i=1}^{m} \in \mathbb{W}$,

$$\{((\boldsymbol{w}, \boldsymbol{X}), (\boldsymbol{w}', \boldsymbol{X}')) \in \mathbb{M} \times \mathbb{M} \mid F(\boldsymbol{w}, \boldsymbol{X}; \boldsymbol{a}, b) = F(\boldsymbol{w}', \boldsymbol{X}'; \boldsymbol{a}, b), \forall (\boldsymbol{a}, b) \in \mathbb{W}\} =$$
$$\{((\boldsymbol{w}, \boldsymbol{X}), (\boldsymbol{w}', \boldsymbol{X}')) \in \mathbb{M} \times \mathbb{M} \mid F(\boldsymbol{w}, \boldsymbol{X}; \boldsymbol{a}_i, b_i) = F(\boldsymbol{w}', \boldsymbol{X}'; \boldsymbol{a}_i, b_i), \forall i = 1, \ldots, m\}. \tag{7}$$

Let $\boldsymbol{A} \in \mathbb{R}^{m \times d}$ with rows $\boldsymbol{a}_1, \ldots, \boldsymbol{a}_m$, and $\boldsymbol{b} = (b_1, \ldots, b_m)$. Suppose that $\boldsymbol{A}, \boldsymbol{b}$ indeed satisfy (7).

Let $\mu, \mu' \in \mathcal{M}_{\leq n}(\Omega)$ be two measures with parameters $(\boldsymbol{w}, \boldsymbol{X})$, $(\boldsymbol{w}', \boldsymbol{X}')$ respectively. Equation (7) implies that if the function $\hat{f}$ of (5) satisfies $\hat{f}(\mu) = \hat{f}(\mu')$, then $(\boldsymbol{w}, \boldsymbol{X})$, $(\boldsymbol{w}', \boldsymbol{X}')$ are not separated by the entire family of functions $\{F(\,\cdot\,; \boldsymbol{a}, b) \mid \boldsymbol{a} \in \mathbb{R}^d, b \in \mathbb{R}\}$. Since $\sigma$ is discriminatory, this in turn implies that $\mu = \mu'$. This concludes the proof of moment injectivity on $\mathcal{M}_{\leq n}(\mathbb{R}^d)$.

If we are only interested in moment injectivity on $\mathcal{S}_{\leq n}(\mathbb{R}^d)$, it is sufficient to apply the theorem to

$$\mathbb{M} = \bigcup_{\boldsymbol{w} \in \{0,1\}^n} \{(\boldsymbol{w}, \boldsymbol{X}) \mid \boldsymbol{X} \in \mathbb{R}^{d \times n}\},$$

which is a finite union of affine subspaces of dimension $D = nd$. Thus, Theorem 3.4 only requires $m = 2nd + 1$ to achieve injectivity on $\mathcal{S}_{\leq n}(\mathbb{R}^d)$. Similarly, when considering $\mathcal{M}_{\leq n}(\Sigma)$ with a countable $\Sigma$, the theorem can be applied to a domain $\mathbb{M}$ that can be written as a countable union of affine spaces of dimension $n$, which yields $m = 2n + 1$. Finally, $\mathcal{S}_{\leq n}(\Sigma)$ is a countable union of points, namely zero-dimensional affine subspaces, and therefore $m = 1$ is sufficient in this case. $\square$

## 3.1 More on the finite witness theorem

The finite witness theorem can be used to prove moment injectivity for functions beyond the activated inner-product form of (5). As an example, we show in the following proposition that Gaussian functions with random parameters are moment injective:

**Proposition 3.5.** *Let $n, d \in \mathbb{N}$ and set $m = 2n(d+1)+1$. Let $\mathbb{W} = (\boldsymbol{y}, \sigma) \in \mathbb{R}^d \times \mathbb{R}_+$. Then for Lebesgue almost any $(\boldsymbol{y}_i, \sigma_i)_{i=1}^{m} \in \mathbb{W}^m$, the function*

$$f(\boldsymbol{x}) = \left( \exp\left( -\frac{\|\boldsymbol{x} - \boldsymbol{y}_1\|^2}{\sigma_1^2} \right), \ldots, \exp\left( -\frac{\|\boldsymbol{x} - \boldsymbol{y}_m\|^2}{\sigma_m^2} \right) \right)$$

*is moment injective on $\mathcal{M}_{\leq n}(\mathbb{R}^d)$.*

*Proof idea.* Any two measures with bounded support can be separated by the moment of a Gaussian function supported on a small ball around a point where the measures disagree. Thus, a measure in $\mathcal{M}_{\leq n}(\Omega)$ is uniquely defined by the continuous family of all its Gaussian moments. The finite witness theorem then shows that a finite number $m$ suffices. $\square$

The full version of the finite witness theorem (Theorem A.2), discussed in Appendix A, is more general than Theorem 3.4. In this version, the class of sets admissible as $\mathbb{M}$ is the class of $\sigma$-subanalytic sets. While its definition is technically involved (see Appendix A), this class is quite vast: it includes all open sets, all semialgebraic sets (including affine spaces, polygons, and closed $\ell_2$-balls), and countable unions thereof. The analyticity assumption on $F$ is also substantially relaxed to $\sigma$-subanalyticity, though this requires an additional condition (13) in the theorem assumptions.

The proof of the finite witness theorem is non-trivial, and we regard it as the main technical contribution of this work. In essence, the proof generalizes a similar result in [9], which only applies to polynomial functions on sets defined by polynomial constraints — known as *semialgebraic sets*. This class of sets has several nice properties, which the proof in [9] relies on: It is closed under linear projections, finite unions, finite intersections, and complements. Moreover, any semialgebraic set is a finite union of smooth manifolds.

Our generalization from the polynomial to the analytic setting consists of two steps: First, we generalize the theorem to a larger class of sets, called *globally subanalytic sets*, which are known to be an *o-minimal system* — essentially, a family of sets that has the same nice properties of semialgebraic sets mentioned above. This generalization is straightforward; however, it does not allow $F$ to be an arbitrary analytic function, and thus does not suffice even to prove the weaker version, Theorem 3.4. Our second step is then to observe that our proof carries through also when considering countable unions of globally subanalytic sets, which we name $\sigma$-*subanalytic sets*. This, in turn, paves the way to prove the full version of the finite witness theorem.

Using the more general version of the theorem, we can prove the following proposition, which in particular implies moment injectivity of *deep* networks, provided that the last activation is analytic:

**Proposition 3.6.** *Let $\sigma : \mathbb{R} \to \mathbb{R}$ be an analytic non-polynomial function. Let $n, d \in \mathbb{N}$ and set $m = 2n(d+1) + 1$. Let $f : \mathbb{R}^d \to \mathbb{R}^L$ be an injective function that is a composition of PwL functions and analytic functions. Then for Lebesgue almost any $\boldsymbol{A} \in \mathbb{R}^{m \times L}, \boldsymbol{b} \in \mathbb{R}^m$, the function $\sigma(\boldsymbol{A}f(\boldsymbol{x}) + \boldsymbol{b})$ is moment injective on $\mathcal{M}_{\leq n}(\mathbb{R}^d)$.*

In particular, $F$ could be a neural network that has increasing widths, linear layers with full rank, and injective activations that are either PwL or analytic (such as leaky ReLU or sigmoid). Therefore, Proposition 3.6 shows that increasing the network depth will not have a negative effect on its moment-injectivity. While this may seem trivial, what is not immediate in this formulation is that the embedding dimension $m$ depends linearly on $n \cdot d$ rather than $n \cdot L$. The reason this is true is that the shallow neural network applied to $F(\boldsymbol{x})$ will only 'see' inputs that originate from the set $F(\mathbb{R}^d)$, and in Appendix A we show that this is a $\sigma$-subanalytic set of dimension $\leq d$.

## 4 Failure of moment injectivity for piecewise-linear functions

In this section, we show that moments of neural networks with piecewise-linear activations (such as ReLU, leaky ReLU and the hard hyperbolic tangent) cannot be injective when the alphabet is infinite, except for some singular cases discussed below.

**Proposition 4.1.** *Let $d, m$ and $n \geq 2$ be natural numbers and $\Omega \subseteq \mathbb{R}^d$ an open set. If $\boldsymbol{\psi} : \mathbb{R}^d \to \mathbb{R}^m$ is piecewise linear, then it is not moment injective on $\mathcal{S}_{\leq n}(\Omega)$.*

*Proof.* There exists some open $U \subset \mathbb{R}^d$ such that $\boldsymbol{\psi}(\boldsymbol{x})$ is of the form $\boldsymbol{\psi}(\boldsymbol{x}) = \boldsymbol{A}\boldsymbol{x} + \boldsymbol{b}$ in $U$. Let $\boldsymbol{x}_0 \in U$ and let $\boldsymbol{d} \neq \boldsymbol{0} \in \mathbb{R}^d$. For small enough $\epsilon > 0$, we have that $\boldsymbol{x}_0 + \epsilon\boldsymbol{d}$ and $\boldsymbol{x}_0 - \epsilon\boldsymbol{d}$ are in $U$. It follows that the multisets $\{\!\{\boldsymbol{x}_0, \boldsymbol{x}_0\}\!\}$ and $\{\!\{\boldsymbol{x}_0 - \epsilon\boldsymbol{d}, \boldsymbol{x}_0 + \epsilon\boldsymbol{d}\}\!\}$ have the same moments:

$$\boldsymbol{\psi}(\boldsymbol{x}_0) + \boldsymbol{\psi}(\boldsymbol{x}_0) = 2(\boldsymbol{A}\boldsymbol{x}_0 + \boldsymbol{b}) = \boldsymbol{A}(\boldsymbol{x}_0 - \epsilon\boldsymbol{d}) + \boldsymbol{b} + \boldsymbol{A}(\boldsymbol{x}_0 + \epsilon\boldsymbol{d}) + \boldsymbol{b} = \boldsymbol{\psi}(\boldsymbol{x}_0 - \epsilon\boldsymbol{d}) + \boldsymbol{\psi}(\boldsymbol{x}_0 + \epsilon\boldsymbol{d}).$$

This proves that $\boldsymbol{\psi}$ is not moment injective on $\mathcal{S}_{\leq n}(\Omega)$. $\qquad\square$

The basic idea behind the above proof is that inside a linear region of $\boldsymbol{\psi}$, different multisets with the same center of mass have the same moments. The same idea can be used to prove failure of moment injectivity of PwL functions on $\mathcal{M}_{\leq n}(\Omega)$ for *any* infinite $\Omega$, and on $\mathcal{S}_{\leq n}(\mathbb{Z}^d)$. On the other hand, PwL networks can be moment injective on $\mathcal{M}_{\leq n}(\Omega)$ with finite $\Omega$, as well as on $\mathcal{S}_{\leq n}(\Sigma)$ when $\Sigma$ is a somewhat pathological infinite countable alphabet. These results are described in Appendix B.

# 5    Failure of bi-Lipschitzness for general moment functions

In Section 3 we have shown that a neural network $f : \mathbb{R}^d \to \mathbb{R}^m$ with analytic non-polynomial activation can induce an injective multiset function $\hat{f} : \mathcal{S}_{\leq n}(\mathbb{R}^d) \to \mathbb{R}^m$. Ideally, we wish such $\hat{f}$ to be *bi-Lipschitz*, meaning that there exist constants $0 < c \leq C$ such that

$$c \cdot W_2(S_1, S_2) \leq \| \hat{f}(S_1) - \hat{f}(S_2) \| \leq C \cdot W_2(S_1, S_2), \quad \forall S_1, S_2 \in \mathcal{S}_{\leq n}(\mathbb{R}^d), \tag{8}$$

where $W_2(S_1, S_2)$ is the 2-Wasserstein distance between the two measures $\mu_1, \mu_2$ that assign uniform weights to the points in $S_1, S_2$ respectively. Unfortunately, we find that *any* moment function $\hat{f}$ induced by some $f : \mathbb{R}^d \to \mathbb{R}^m$ cannot be bi-Lipschitz, assuming that $f$ is differentiable in at least one point.

**Proposition 5.1.** *Let $n \geq 2$, $d, m \in \mathbb{N}$, and let $f : \mathbb{R}^d \to \mathbb{R}^m$ be differentiable at some $\boldsymbol{x}_0 \in \mathbb{R}^d$. Then the induced moment function $\hat{f} : \mathcal{S}_{\leq n}(\mathbb{R}^d) \to \mathbb{R}^m$ defined in (1) is not bi-Lipschitz.*

Figure 1(b) illustrates the underlying reason for this failure of bi-Lipschitzness, and its relation to the non-injectivity of PwL moments: consider a shallow neural network $f : \mathbb{R} \to \mathbb{R}^{10}$ with ReLU activations, and its induced moment function $\hat{f}(\{\!\{x_1, x_2\}\!\}) = f(x_1) + f(x_2)$ on multisets in $\mathcal{S}_{\leq 2}(\mathbb{R})$. The left-hand side visualizes the ratio $\sigma_2/\sigma_1$ of the smallest and largest singular values of the differential matrix $D\hat{f}$. The function $f$ is PwL, with four linear regions $I_1, \ldots, I_4$ in $[0, 1]$. The linear regions of $\hat{f}$ in $[0, 1]^2$ are thus the rectangles $I_i \times I_j$. As seen in the figure, there are degeneracies in the rectangles that intersect the diagonal, as for small enough $\epsilon$, $\hat{f}(\{\!\{x_0 + \epsilon, x_0 - \epsilon\}\!\}) = \hat{f}(\{\!\{x_0, x_0\}\!\})$ as in the proof of Proposition 4.1. The right-hand side visualizes the same ratio when the analytic SiLU activation is used instead of ReLU. We see that $D\hat{f}$ is singular on the diagonal. Intuitively, this is because the differentiability of $f$ implies that it behaves locally like an affine function. This leads to singularities of $\hat{f}$ on the diagonal, which do not prevent it from being injective, but do prevent it from being bi-Lipschitz. A proof of this phenomenon is given in the appendix. See also Theorem 21 in [4], which independently proved a similar result for general invariant embeddings.

# 6    Applications: Universal Approximation and Graph Separation

As mentioned in the introduction, injective multiset functions can be used to construct multiset architectures with universal approximation power, and to prove separation results for graph neural networks. In this section, we present some immediate corollaries of our results for these two applications. Proofs are in Appendix D.3.

## 6.1    Universal approximation of functions on multisets and measures

Our first approximation result focuses on multisets of a fixed size $n$ with an alphabet $K \subseteq \mathbb{R}^d$ that is compact. Any such multiset is determined by a choice of $n$ vectors in $K$, possibly with repetitions and irrespective of order. Thus, multiset functions on this space are equivalent to permutation-invariant functions on $K^n$. Using the finite witness theorem and a basic topological argument, we prove:

**Corollary 6.1.** *Let $n, d \in \mathbb{N}$ and set $m = 2nd + 1$. Let $\sigma : \mathbb{R} \to \mathbb{R}$ be an analytic non-polynomial function. Let $K \subseteq \mathbb{R}^d$ be a compact set. Then there exist $\boldsymbol{A} \in \mathbb{R}^{m \times d}, \boldsymbol{b} \in \mathbb{R}^d$ such that for any continuous permutation-invariant $f : K^n \to \mathbb{R}$, there exists a continuous $F : \mathbb{R}^m \to \mathbb{R}$ such that*

$$f(\boldsymbol{X}) = F\left( \sum_{j=1}^n \sigma(\boldsymbol{A}\boldsymbol{x}_j + \boldsymbol{b}) \right), \quad \forall \boldsymbol{X} = (\boldsymbol{x}_1, \ldots, \boldsymbol{x}_n) \in K^n. \tag{9}$$

Combining Corollary 6.1 with the universality of MLPs, we get that any continuous permutation-invariant function on $K^n$ can be approximated by expressions of the form (9) with $F$ replaced by an MLP. Similar results were obtained for moments of polynomials rather than of MLPs in [45, 42, 9].

It is worth noting that an analogue of Corollary 6.1 cannot hold with a piecewise-linear $\sigma$, assuming that $K$ has a non-empty interior. This is because by Proposition 4.1, any fixed moment function induced by a PwL MLP will not be able to separate all multisets, whereas any two distinct multisets can be separated by some continuous $f$. Though, with a PwL $\sigma$, one may approximate any given $f$ to

arbitrary precision, by taking the embedding dimension $m$ to infinity. In contrast, with an analytic $\sigma$, we are able to specify a *finite* $m = 2nd + 1$ for which exact equality in (9) is guaranteed.

Since our injectivity results on multisets extend to measures, it is natural to seek an extension of the above approximation result to functions defined on measures. Denote by $\mathcal{P}_{\leq n}(K)$ the space of probability measures supported on $\leq n$ points in $K \subseteq \mathbb{R}^d$, endowed with the 2-Wasserstein metric.

**Corollary 6.2.** *Let $n, d \in \mathbb{N}$ and set $m = 2n(d + 1) + 1$. Let $\sigma : \mathbb{R} \to \mathbb{R}$ be analytic and non-polynomial. Let $K \subseteq \mathbb{R}^d$ be compact. Then there exist $\boldsymbol{A} \in \mathbb{R}^{m \times d}, \boldsymbol{b} \in \mathbb{R}^m$ such that for any continuous (in the 2-Wasserstein sense) $f : \mathcal{P}_{\leq n}(K) \to \mathbb{R}$, there exists a continuous $F : \mathbb{R}^m \to \mathbb{R}$ such that*

$$f(\mu) = F\left( \int_{\boldsymbol{x} \in K} \sigma(\boldsymbol{A}\boldsymbol{x} + \boldsymbol{b}) d\mu(\boldsymbol{x}) \right), \quad \forall \mu \in \mathcal{P}_{\leq n}(K).$$

It follows from Corollary 6.2 that any continuous function $f : \mathcal{P}_{\leq n}(K) \to \mathbb{R}$ with compact $K \subseteq \mathbb{R}^d$ can be approximated to arbitrary precision by functions of the form $\hat{f}(\mu) = F\left( \int_{\boldsymbol{x} \in K} \sigma(\boldsymbol{A}\boldsymbol{x} + \boldsymbol{b}) d\mu(\boldsymbol{x}) \right)$, with $\boldsymbol{A} \in \mathbb{R}^{m \times d}, \boldsymbol{b} \in \mathbb{R}^m$, $m = 2nd + 1$, and $F$ being an MLP.

## 6.2 Graph separation

We now discuss the implications of Theorem 3.3 for graph separation, using terminology from [44]. Let $\mathcal{G}_{\leq n}(\Sigma)$ be the collection of all graphs $G = (V, E, \boldsymbol{h}^{(0)})$ with at most $n$ vertices, endowed with vertex features $\boldsymbol{h}_v^{(0)} \in \Sigma$, where $\Sigma \subseteq \mathbb{R}^d$ is a countable alphabet. We consider GIN-like [44] MPNNs that recursively, for $t = 1, \ldots, T$, calculate node features $\boldsymbol{h}_v^{(t)}$ from the previous features $\boldsymbol{h}_v^{(t-1)}$ by

$$\boldsymbol{h}_v^{(t)} = \sum_{u \in \mathcal{N}(v)} \sigma\left( \boldsymbol{A}^{(t)} \left( \eta^{(t)} \boldsymbol{h}_v^{(t-1)} + \boldsymbol{h}_u^{(t-1)} \right) + \boldsymbol{b}^{(t)} \right). \tag{10}$$

After the $T$ iterations are concluded, a global feature is computed via a *readout function*:

$$\boldsymbol{h}_G = \sum_{v \in V} \sigma\left( \boldsymbol{A}^{(T+1)} \boldsymbol{h}_v^{(T)} + \boldsymbol{b}^{(T+1)} \right). \tag{11}$$

We choose all features $\boldsymbol{h}_G$ and $\boldsymbol{h}_v^{(t)}$ for $1 \leq t \leq T$, to have the same dimension $m$. Based on the fact that MPNNs are equivalent to 1-WL when the multiset functions are injective [44], and on our injectivity results for countable alphabets, we prove that:

**Theorem 6.3.** *Let $n, d, T \in \mathbb{N}$ and let $\Sigma \subseteq \mathbb{R}^d$ be countable. Let $m \geq 1$ be any integer. Let $\sigma : \mathbb{R} \to \mathbb{R}$ be an analytic non-polynomial function. Then for Lebesgue almost any choice of $\boldsymbol{A}^{(t)}, \boldsymbol{b}^{(t)}$ and $\eta^{(t)}$, the MPNN defined in (10) and (11) assigns different global features to any pair of graphs $G_1, G_2 \in \mathcal{G}_{\leq n}(\Sigma)$ that can be separated by $T$ iterations of 1-WL.*

**Graph separation with continuous features** Up to now, we have discussed graphs with node features coming from a countable alphabet. Since Theorem 3.3 applies to multisets with continuous alphabets, it can be applied to separation of graphs with continuous node features as well. In particular, the paper [15] explains how the random semialgebraic multiset function from [9] can be used to construct architectures for graphs with continuous features, whose separation power is equivalent to WL tests. Their focus is on showing that the embedding dimension in their construction depends linearly on the dimension of the *graph space*, rather than grows exponentially with the number of message-passing iterations $T$. Similar results can now be obtained by using our random analytic multiset functions. We leave a full description of these aspects to future work.

## 7 Experiments

**Empirical injectivity and bi-Lipschitzness** We empirically investigated the injectivity and bi-Lipschitzness of moments of shallow networks of the form (3), by randomly generating a large number of pairs of multisets of $n$ vectors in $\mathbb{R}^d$, and computing the optimal constants $c, C$ for which (8) holds for the generated pairs. The ratio $c/C$ for varying activations and embedding dimension $m$ is shown in Figure 2. Here $d = 3$ and $n = 1000$. Similar qualitative results were obtained for other values of $d$ and $n$; see Figure 4 in Appendix E.

We observe several interesting phenomena: First, at low embedding-dimensions, the ratio $c/C$ for PwL networks is exactly zero, indicating that they are not injective even on the finite sample set. In contrast, for analytic activations, $c/C$ is always positive. Indeed, we expect analytic activations to be injective on a finite sample even with embedding dimension $m = 1$, since a finite sample set has an intrinsic dimension of zero. Next, we observe that $c/C$ naturally improves as $m$ increases. Finally, we note that even for high $m$, $c/C$ is rather small.

Indeed, if it were possible to consider *all* pairs of multisets when computing $c/C$, we would get zero for all activations and all embedding dimensions, as follows from Theorem 5.1. For additional details on this experiment, see Appendix E.1.

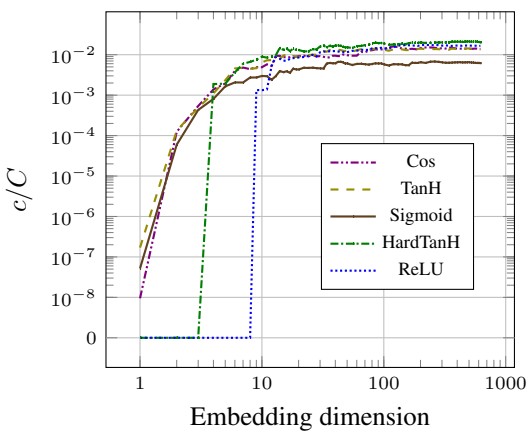

**Figure 2**

**Graph Separation**  To validate Theorem 6.3, we conducted the following experiment: we considered 600 graphs from the TUDataset [30]. On each graph we ran three iterations of the WL test, and three iterations of MPNNs with the Graph Convolutional Layers from [18] with different activations and hidden dimensions. Our goal was to check in how many graphs the MPNNs returned a vertex coloring that differs from the coloring provided by the WL test. The results are shown in Figure 1(a). As seen in the table, with the three *analytic* activations tested, the vertex coloring of MPNN was always equivalent to 1-WL, even with a hidden dimension of 1. On the other hand, for the three *PwL* activations, there were inconsistencies in about 1% of the graphs, even with a hidden dimension of 50.

We note that while analytic activations fully succeeded in separation, in some cases the separation was rather weak: while the distance between features of non-equivalent nodes computed by the MPNNs was typically around $0.1$, the least-separated features had a distance of $\sim 10^{-7}$. In future work, it could be interesting to investigate whether MPNNs can be trained to yield larger distances between the features of all non-equivalent nodes. Further details on this experiment appear in Appendix E.2.

## 8   Conclusion

We have shown that moments of neural networks with an analytic non-polynomial activation are injective on multisets and measures. We have also shown how this can be harnessed to construct universal approximators for multiset functions, as well as prove separation results for graph neural networks. A key advantage of our approach is that it enables constructing proofs using real models that are used in practice, rather than idealized versions of them as done in previous works.

It may seem tempting, due to our theoretical results, to conclude that analytic activations should perform on multisets better than piecewise-linear activations. We stress that we make no such claim. Indeed, while the separation results in Figure 1(a) corroborate our theory, PwL networks fail to separate only $1\%$ of the graphs in our experiment. Furthermore, at high embedding dimensions, the empirical bi-Lipschitzness in Figure 2 does not seem to strongly depend on the analyticity of the activation function. Our claim is thus much more modest: we claim that multiset architectures with analytic activations are easier to *theoretically* analyze, and we hope that pursuing this analysis shall lead to fruitful theoretical and practical insights, which may ultimately benefit multiset architectures with either type of activation.

Lastly, we note that the finite witness theorem, which is presented here as a tool for proving moment injectivity, may prove valuable as a general tool for reducing an infinite number of equality constraints to a finite number, and we believe it will find additional applications beyond the scope of this paper.

**Acknowledgements** N.D. is partially funded by a Horev Fellowship. T.A, R.R. and N.D. are partially funded by ISF grant 272/23.

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

# A   Finite witness theorem

In this section, we state and prove the full version of the finite witness theorem (Theorem A.2), which is more general than Theorem 3.4 stated in the main text. Before laying out the formal definitions and proofs, we begin by describing the context of these results. While this section makes use of notions from algebraic geometry and real analytical functions, it is self-contained and most of it only requires knowledge of elementary calculus and some topology.

The finite witness theorem is essentially a tool for reducing an infinite, continuously parameterized family of constraints $p(z; \theta) = 0$ $\forall \theta$ to a finite subset of constraints $p(z; \theta_i) = 0$, $i = 1, \ldots, m$, defined by random parameters $\theta_1, \ldots, \theta_m$. This general approach seems to have originated from the famous proof of uniqueness for phase-retrieval measurements in [2]. In that work, functions $p(z, \theta) : \mathbb{C}^n \times \mathbb{C}^n \to \mathbb{R}$ of the form $p(z; \theta) = |\langle z, \theta \rangle|$ were considered, and it was proved that a finite number of $\sim 4n$ random measurements $\theta_i$ are sufficient to uniquely determine a signal, up to unavoidable global phase ambiguity. The proof in [2] achieves this result by showing that the values $p(z; \theta)$ for *all* possible $\theta$s are sufficient to determine the signal uniquely, and then uses a real algebraic-geometric and dimension-counting argument to show that this continuous family of measurements can be replaced by a finite subset, defined by $m \sim 4n$ random vectors (*witnesses*) $\{\theta_i\}_{i=1}^m$, without losing information.

In [9], the authors provide a generalization of the results in [2], by defining conditions under which a continuously parameterized family of functions $p(z; \theta)$ that fully determines $z$ up to equivalence, can be replaced by a finite subset $p(z; \theta_i)$, $i = 1, \ldots, m$ determined by random witness-vectors $\theta_i$. This theorem is based on similar arguments as those used in [2], and on similar assumptions required for machinery from real algebraic geometry. Specifically, sets in [9] are assumed to be *semialgebraic*, which means that they are finite unions of subsets of $\mathbb{R}^D$ that can be defined by polynomial equalities and inequalities. This class of sets includes, for example, finite unions of spheres, balls, and convex polyhedra. The functions in the theorem are assumed to be semialgebraic as well, which means that their graphs are semialgebraic sets. This class of functions includes polynomials, as well as rational functions and piecewise-linear functions.

The main theorem in [9] can be essentially[3] formulated as

**Theorem A.1.** *Let $\mathbb{M}$ be a semialgebraic set of dimension $D$, and let $F : \mathbb{M} \times \mathbb{R}^{D_\theta} \to \mathbb{R}$ be a semialgebraic function. Define the set*

$$\mathcal{N} = \{z \in \mathbb{M} \mid F(z; \theta) = 0, \ \forall \theta \in \mathbb{R}^{D_\theta}\}$$

*and assume that for all $z \in \mathbb{M} \setminus \mathcal{N}$, we have that*

$$\dim\{\theta \in \mathbb{R}^{D_\theta} \mid F(z; \theta) = 0\} \le D_\theta - 1, \tag{12}$$

*then for Lebesgue almost every $\theta^{(1)}, \ldots, \theta^{(D+1)}$,*

$$\mathcal{N} = \{z \in \mathbb{M} \mid F(z; \theta^{(i)}) = 0, \ \forall i = 1, \ldots D + 1\}.$$

The notion of dimension used in condition (12) and throughout this section is the *Hausdorff dimension*, explained in Appendix A.4 below.

Theorem A.1 is similar to the simple version of the finite witness theorem (Theorem 3.4), with four notable differences: (1) The domain $\mathbb{M}$ in Theorem 3.4 is a countable union of affine sets; this is not, in general, a semialgebraic set, and thus does not qualify for the conditions of Theorem A.1. (2) The function $F$ in Theorem 3.4 is analytic. Analytic functions are not necessarily semialgebraic. (3) Theorem A.1 requires the extra condition (12), which does not appear in Theorem 3.4. This condition is not required in Theorem 3.4 because with analytic functions, it is always satisfied; however, it will be required in our full version of the theorem, since it admits a more general class of functions. (4) Theorem A.1 deals with sets of the form $\{z \mid F(z, \theta) = 0, \forall \theta\}$ while Theorem 3.4 deals with sets of the form $\{(x, y) \mid F(x, \theta) = F(y, \theta), \forall \theta\}$. This difference is not essential and can be handled by the change of variables $z = (x, y)$ and $\tilde{F}(x, y, \theta) = F(x, \theta) - F(y, \theta)$.

---

[3]To obtain this theorem from the formulation in [9], use the change of variables $z = (x, y)$ and $F(z; \theta) = p(x; \theta) - p(y; \theta)$. The formulation in [9] makes some additional requirements on $F$ which are relevant to the specific applications considered there, but going through the details of the proof shows that it is sufficient for proving Theorem A.1 as stated here.

To address the first two differences, we shall generalize Theorem A.1 to support a large class of domains $\mathbb{M}$ and functions $F$. The admissible domains shall include a vast class of sets, among which are *countable* unions of semialgebraic sets, as well as all open sets, and sets defined by analytic, rather than polynomial, equations. The admissible functions $F$ shall include semialgebraic functions, analytic functions, and many other types of functions. To achieve this goal, we use results from the study of *o-minimal structures* (see below), which aim at finding families of sets that have the same tame properties as semialgebraic sets.

A good starting point to achieve this generalization is to consider the family of *globally subanalytic sets* (formally defined below). This family contains all semialgebraic sets and is an o-minimal system; consequently, it is possible to generalize Theorem A.1 so that the domain $\mathbb{M}$ can include all globally subanalytic sets, and the function $F$ could be any *globally subanalytic function* (meaning that the graph of $F$ is a globally subanalytic set). However, this still will not allow for a general analytic $F$, nor for $\mathbb{M}$ to include countable unions of affine sets.

To address this, we will show that an analogue to Theorem A.1 holds even when considering *countable unions* of globally subanalytic sets. We call such sets $\sigma$-*subanalytic sets*. To the best of our knowledge, such sets have not been studied to date. The family of $\sigma$-subanalytic sets is not an o-minimal structure, since it not closed under taking complements. However, it *is* closed under linear projections, countable unions, finite intersections and Cartesian products. Moreover, $\sigma$-subanalytic sets are countable unions of $C^\infty$ manifolds. We find that these properties are sufficient to generalize Theorem A.1 and obtain a finite witness theorem for the $\sigma$-subanalytic category.

Clearly, the class of $\sigma$-subanalytic sets is larger than the class of globally subanalytic sets. In particular, it includes countable unions of semialgebraic sets. This enables the domain $\mathbb{M}$ in the theorem statement to be a countable union of affine spaces, as in Theorem 3.4 from the main text. This also implies that any open set is $\sigma$-subanalytic, since it is a countable union of open balls — which are semialgebraic sets.

As for the function $F$, our theorem admits all $\sigma$-subanalytic functions: functions whose graph is $\sigma$-subanalytic. This class includes all analytic functions, as well as all semialgebraic functions. Moreover, we show below that it is closed under composition and other elementary operations.

We now state the full version of the finite witness theorem.

**Theorem A.2** (Finite Witness Theorem, full version). *Let $\mathbb{M} \subseteq \mathbb{R}^p$, $\mathbb{W} \subseteq \mathbb{R}^q$ be $\sigma$-subanalytic sets of dimension $D$ and $D_{\boldsymbol\theta}$ respectively. Let $F : \mathbb{M} \times \mathbb{W} \to \mathbb{R}$ be a $\sigma$-subanalytic function. Define the set*

$$\mathcal{N} = \{\boldsymbol{z} \in \mathbb{M} \mid F(\boldsymbol{z}; \boldsymbol{\theta}) = 0, \ \forall \boldsymbol{\theta} \in \mathbb{W}\}.$$

*Suppose that for all $\boldsymbol{z} \in \mathbb{M} \setminus \mathcal{N}$*

$$\dim\{\boldsymbol{\theta} \in \mathbb{W} \mid F(\boldsymbol{z}; \boldsymbol{\theta}) = 0\} \leq D_{\boldsymbol\theta} - 1. \tag{13}$$

*Then for generic $\left(\boldsymbol{\theta}^{(1)}, \ldots, \boldsymbol{\theta}^{(D+1)}\right) \in \mathbb{W}^{D+1}$,*

$$\mathcal{N} = \{\boldsymbol{z} \in \mathbb{M} \mid F(\boldsymbol{z}; \boldsymbol{\theta}^{(i)}) = 0, \ \forall i = 1, \ldots D + 1\}. \tag{14}$$

*Moreover, if $\mathbb{W}$ is an open and connected subset of $\mathbb{R}^q$, and $F(\boldsymbol{z}; \boldsymbol{\theta})$ is analytic as a function of $\boldsymbol{\theta}$ for all fixed $\boldsymbol{z} \in \mathbb{M}$, then condition (13) is not required, as it is automatically satisfied.*

The notion of dimension in (13) is the Hausdorff dimension (discussed below), and the term *generic* means that the set of $\left(\boldsymbol{\theta}^{(1)}, \ldots, \boldsymbol{\theta}^{(D+1)}\right)$ for which (14) fails is a subset of $\mathbb{W}^{D+1}$ whose Hausdorff dimension is strictly lower than $\dim\left(\mathbb{W}^{D+1}\right)$. In particular, in the common case where $\mathbb{W} = \mathbb{R}^q$, or is an open subset of $\mathbb{R}^q$, we have that the theorem holds for almost any $\left(\boldsymbol{\theta}^{(1)}, \ldots, \boldsymbol{\theta}^{(D+1)}\right)$, with respect to the Lebesgue measure on $\mathbb{R}^{q(D+1)}$ .

We now begin to rigorously define the notions discussed in this section and then prove Theorem A.2. In Appendices A.1 to A.4, we construct the theoretical framework step by step. Then, in Appendix A.5 we prove the theorem, and in Appendix A.6 we show that the simple version of the theorem (Theorem 3.4) follows from the full version. Finally, we present in Appendix A.6 several corollaries to the theorem.

## A.1 Definitions and Background

### A.1.1 Globally subanalytic sets: an o-minimal structure

We begin by defining analytic functions, subanalytic sets and functions, and various other related concepts that will be required for our discussion. These definitions are taken from [19] and [38].

**Definition A.3** (Analytic function [19])**.** Let $U \subseteq \mathbb{R}^D$ be an open set. We say that $f : U \to \mathbb{R}$ is *analytic* if for all $z \in U$, there exists an open ball $V \subseteq U$ centered at $z$, and $(a_\alpha)_{\alpha \in \mathbb{N}^D}$ such that

$$f(\boldsymbol{y}) = \sum_{\alpha \in \mathbb{N}^D} a_\alpha (\boldsymbol{y} - \boldsymbol{z})^\alpha \quad \forall \boldsymbol{y} \in V, \tag{15}$$

and the power series in (15) converges absolutely.

Our next goal is to define the family of globally subanalytic sets, which contains all semialgebraic sets; moreover, it is an *o-minimal structure*, meaning that it shares the essential tame properties of semialgebraic sets. This requires some preliminary definitions:

**Definition A.4** (Semianalytic sets [38])**.** A subset $E \subseteq \mathbb{R}^n$ is called *semianalytic* if it is locally defined by finitely many real analytic equalities and inequalities. Namely, for each $a \in \mathbb{R}^n$, there is a neighborhood $U$ of $a$, and real analytic functions $f_{ij}, g_{ij}$ on $U$, where $i = 1, \dots, r$ and $j = 1 \dots s_i$, such that

$$E \cap U = \bigcup_{i=1}^{r} \bigcap_{j=1}^{s_i} \{\boldsymbol{z} \in U \,|\, g_{ij}(\boldsymbol{z}) > 0 \text{ and } f_{ij}(\boldsymbol{z}) = 0\}. \tag{16}$$

**Example A.5.** As shown in Example 1.1.2 in [38], the graph of the analytic function $f : (0,1) \to \mathbb{R}$ defined by $f(x) = \sin(1/x)$ is *not* a semianalytic set. This is because there is no neighborhood of $a = (0,0)$ for which (16) holds. On the other hand, if $B \subseteq \mathbb{R}^n$ is a closed ball and $f : B \to \mathbb{R}$ can be extended to an analytic function in an open set containing $B$, then the graph of $f$ (as a function defined on $B$) will be semianalytic.

The example above suggests that the class of semianalytic sets may be too restrictive for our purposes. More importantly, linear projections of semianalytic sets may not be semianalytic [38], so the family of semianalytic sets does not form an o-minimal system. This can be remedied by defining globally semianalytic sets (which don't form an o-minimal system) and then using these sets to define globally subanalytic sets (which *do* form an o-minimal system):

**Definition A.6** (Globally semianalytic sets [38])**.** A subset $Z \subseteq \mathbb{R}^n$ is *globally semianalytic* if $V_n(Z)$ is a semianalytic subset of $\mathbb{R}^n$, where $V_n : \mathbb{R}^n \to (-1,1)^n$ is the homeomorphism defined by

$$V_n\left(\boldsymbol{z} = (z_1, \dots, z_n)\right) = \left( \frac{z_1}{\sqrt{1 + |\boldsymbol{z}|^2}}, \dots, \frac{z_n}{\sqrt{1 + |\boldsymbol{z}|^2}} \right).$$

Globally semianalytic sets can be thought of as semianalytic sets that remain semianalytic when 'compactified' by $V_n$. This is useful to rule out bad behaviour as $\boldsymbol{z}$ goes out to infinity. Important examples of globally semianalytic sets include all bounded semianalytic sets and all semialgebraic sets [38].

Globally semianalytic sets are still not closed under linear projection. Their projections are called globally *subanalytic* sets:

**Definition A.7** (Globally subanalytic sets [38])**.** A subset $E \subseteq \mathbb{R}^n$ is *globally subanalytic* if it can be presented as a linear projection of a globally semianalytic set; more precisely, if there exists a globally semianalytic set $Z \subseteq \mathbb{R}^{n+p}$ such that $E = \pi(Z)$, with $\pi : \mathbb{R}^{n+p} \to \mathbb{R}^n$ being the projection operator that omits the last $p$ coordinates while leaving the remaining coordinates unchanged.

Globally semianalytic sets are by definition globally subanalytic. Globally subanalytic sets do form an *o-minimal structure*. In particular, they have the following properties:

**Proposition A.8** (Properties of globally subanalytic sets [38])**.** *Let $A, B \subseteq \mathbb{R}^D$ and $C \subseteq \mathbb{R}^M$ be globally subanalytic sets. Then:*

  *1. $\mathbb{R}^D \setminus A$ is globally subanalytic.*

2. $A \cup B$ is globally subanalytic.

3. $A \cap B$ is globally subanalytic.

4. $A \times C$ is globally subanalytic.

5. If $\pi : \mathbb{R}^D \to \mathbb{R}^L$ is a linear projection, then $\pi(A)$ is globally subanalytic.

6. $A$ is a finite union of $C^\infty$ manifolds.

The first property in Proposition A.8 is proven in Gabrielov's Complement Theorem; see Theorem 1.8.8 in [38]. The last property is a weaker version of Theorem 1.2.3 in [38]. The remaining properties are proved[4] in [38, Basic properties 1.1.8].

## A.2 $\sigma$-subanalytic sets

As mentioned above, the fact that globally subanalytic sets form an o-minimal structure is already sufficient to prove a finite witness theorem for this class. However, this class is still too restrictive for our purposes, as it does not allow us to support arbitrary analytic functions $F$, as the following example shows:

**Example A.9.** Analogously to semialgebraic functions, we say that a function is *globally subanalytic* if its graph is globally subanalytic. Consider the function $\sin(x)$ defined on the real line. Then by Example 1.1.7 in [38], this function is not globally subanalytic.

Fortunately, we find that a finite witness theorem can be proven for a larger class of sets: countable unions of globally subanalytic sets. We name such sets $\sigma$-*subanalytic* sets. To the best of our knowledge, this class of sets has not been studied to date[5]. The class of $\sigma$-subanalytic sets is rather large: we will show below that it contains all open sets, while the classes discussed previously do not (see Example A.12 below). This is illustrated in the Venn diagram on the left-hand side of Figure 3. Correspondingly, the class of $\sigma$-subanalytic functions is larger than the classes of functions considered previously, and in particular it contains *any* analytic function $F$ defined on *any* open set. This is illustrated in the Venn diagram on the right-hand side of Figure 3, and will be discussed rigorously in Appendix A.3 below. Thus, moving to $\sigma$-*subanalytic* sets and functions is the crucial step that enables us to achieve our goal of proving a finite witness theorem for analytic functions (though the theorem covers a much larger class).

We now define $\sigma$-subanalytic sets and study their properties.

**Definition A.10** ($\sigma$-subanalytic sets)**.** We say that a subset $A \subseteq \mathbb{R}^D$ is $\sigma$-*subanalytic* if it is a countable union of globally subanalytic subsets of $\mathbb{R}^D$.

The class of $\sigma$-subanalytic sets is rather large. For example, any open set in $\mathbb{R}^D$ can be written as a countable union of open balls. Since open balls are semialgebraic sets, it follows that all open sets are $\sigma$-subanalytic. Also, any semianalytic set is $\sigma$-subanalytic, since semianalytic sets are a countable union of bounded semianalytic sets, and these are globally subanalytic.

### Properties of $\sigma$-subanalytic sets

As the following proposition shows, $\sigma$-subanalytic sets inherit most of the properties enjoyed by globally subanalytic sets, described in Proposition A.8.

**Proposition A.11** (Properties of $\sigma$-subanalytic sets)**.** *Assume that $A, B \subseteq \mathbb{R}^D$ and $C \subseteq \mathbb{R}^M$ are $\sigma$-subanalytic sets, then*

1. *$A \cup B$ is $\sigma$-subanalytic. More generally, any countable union of $\sigma$-subanalytic sets is $\sigma$-subanalytic.*

---

[4]Regarding the fifth property in Proposition A.8, note that it does not follow directly from property 1 in [38] which only discusses a special class of projections, but rather from property 4 in [38] which states that the image of a globally subanalytic set under a globally subanalytic mapping is globally subanalytic. Any linear projection is a semialgebraic mapping, and hence a globally subanalytic mapping.

[5]Model theorists and analytic geometers *have* researched other structures that are larger than o-minimal structures and enjoy some of their tame properties; see, e.g., [39, 27].

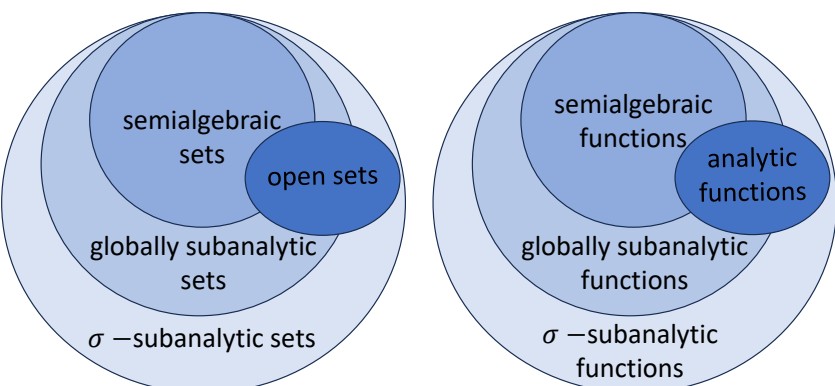

**Figure 3:** Left: The class of semialgebraic sets is contained in the class of globally subanalytic sets, which in turn is contained in the class of $\sigma$-subanalytic sets — on which we focus in this paper. One of the advantages of this larger class is that it contains all open sets, whereas the two smaller classes do not. Right: The classes of semialgebraic, globally subanalytic, and $\sigma$-subanalytic functions are related in the same way. The class of $\sigma$-subanalytic functions is the only one of the three that contains all analytic functions.

2. $A \cap B$ is $\sigma$-subanalytic.

3. $A \times C$ is $\sigma$-subanalytic.

4. If $\pi : \mathbb{R}^D \to \mathbb{R}^L$ is a linear projection, then $\pi(A)$ is a $\sigma$-subanalytic set.

5. $A$ is a countable *union of* $C^\infty$ *manifolds.*

*Proof.* Let $A, B \subseteq \mathbb{R}^D$ be $\sigma$-subanalytic sets. Then $A = \cup_{n \in \mathbb{N}} A_n$ and $B = \cup_{m \in \mathbb{N}} B_m$, where each $A_n$ and $B_m$ is globally subanalytic. We then have

1. A countable union of $\sigma$-analytic sets, each of which being itself a countable union of globally subanalytic set, is clearly a countable union of globally subanalytic sets, and therefore is $\sigma$-subanalytic by definition.

2. We have that

$$A \cap B = (\cup_{n \in \mathbb{N}} A_n) \cap (\cup_{m \in \mathbb{N}} B_m) = \cup_{(n,m) \in \mathbb{N}^2} (A_n \cap B_m)$$

and $A_n \cap B_m$ is globally subanalytic as the intersection of globally subanalytic sets.

3. The set $A \times \mathbb{R}^M$ can be presented as

$$A \times \mathbb{R}^M = (\cup_{n \in \mathbb{N}} A_n) \times \mathbb{R}^M = \cup_{n \in \mathbb{N}} \left( A_n \times \mathbb{R}^M \right),$$

with $A_m$ being globally subanalytic. Since globally subanalytic sets are closed to Cartesian products, each $A_n \times \mathbb{R}^M$ is globally subanalytic, and their countable union is $\sigma$-subanalytic. Finally,

$$A \times C = \left( A \times \mathbb{R}^M \right) \cap \left( \mathbb{R}^D \times C \right),$$

which is $\sigma$-subanalytic by part 2.

4. Follows from globally subanalytic sets being closed to projections, since

$$\pi \left( \cup_{n \in \mathbb{N}} A_n \right) = \cup_{n \in \mathbb{N}} \left( \pi(A_n) \right).$$

5. A $\sigma$-subanalytic set is a countable union of globally subanalytic sets, each of which is a finite union of $C^\infty$ manifolds. Therefore, it is a countable union of $C^\infty$ manifolds.

$\square$

In comparison to Proposition A.8, we see that we have lost two properties by moving from globally subanalytic sets to countable unions thereof: Firstly, a $\sigma$-subanalytic set is a countable union of manifolds rather than a finite one. Secondly, the complement of a $\sigma$-subanalytic set may not be $\sigma$-subanalytic, as our next example shows. As it turns out, the properties noted in Proposition A.11 are sufficient for our purposes.

**Example A.12.** The Cantor set is not $\sigma$-subanalytic, since it has fractal Hausdorff dimension and thus is not a countable union of manifolds. However, the complement of the Cantor set, which we denote by $U$, is an open set and hence is $\sigma$-subanalytic. This shows that the class of $\sigma$-subanalytic sets is not closed to taking complements, as well as countable intersections: this is since the Cantor set is a countable intersection of finite unions of intervals, which are $\sigma$-subanalytic. The set $U$ is also an example of an open set that is not semialgebraic or globally subanalytic, since these classes of sets *are* closed to complements, and do not contain the Cantor set.

### A.3 $\sigma$-subanalytic functions

We now define the class of functions our theorem can admit as the function $F$: $\sigma$-subanalytic functions.

**Definition A.13** ($\sigma$-subanalytic function). Let $\mathbb{M} \subseteq \mathbb{R}^D$ be a $\sigma$-subanalytic set. We say that $f : \mathbb{M} \to \mathbb{R}^L$ is a $\sigma$-subanalytic function if its graph is a $\sigma$-subanalytic subset of $\mathbb{R}^{D+L}$.

If $\mathbb{M} \subseteq \mathbb{R}^D$ is a semialgebraic set and $f : \mathbb{M} \to \mathbb{R}^L$ is a semialgebraic function, then the graph of $f$ is semialgebraic, and thus is $\sigma$-subanalytic. Therefore, a semialgebraic function $f$ defined on a semialgebraic domain $\mathbb{M}$ is a $\sigma$-subanalytic function. A similar result holds also for analytic functions defined on open sets, as shown in the following lemma:

**Lemma A.14.** *If $U \subseteq \mathbb{R}^D$ is open and $f : U \to \mathbb{R}^L$ is analytic, then $f$ is $\sigma$-subanalytic.*

*Proof.* First recall that since $U$ is open, it is a $\sigma$-subanalytic set. Next, we can write $U$ as a countable union of closed balls $B_i$, and then the graph of $f$ can be written as a countable union of sets of the form

$$\{(\boldsymbol{x}, \boldsymbol{y}) \in B_i \times \mathbb{R}^L \,|\, \boldsymbol{y} = f(\boldsymbol{x})\}.$$

These subsets of $U \times \mathbb{R}^L$ are compact and semianalytic, and hence they are globally subanalytic. $\quad\square$

### Properties of $\sigma$-subanalytic functions

The following proposition shows that the class of $\sigma$-subanalytic functions is closed under composition and elementary arithmetic operations. In particular, this allows for functions that combine analytic and semialgebraic functions via compositions and elementary arithmetic operations.

**Proposition A.15.** *Let $A \subseteq \mathbb{R}^a$ and $B \subseteq \mathbb{R}^b$ be $\sigma$-subanalytic sets.*

1. *(Composition) If $f : A \to B$ and $g : B \to \mathbb{R}^c$ are $\sigma$-subanalytic functions, then $g \circ f$ is a $\sigma$-subanalytic function.*

2. *(Concatenation) If $f : A \to \mathbb{R}^m$ and $g : A \to \mathbb{R}^n$ are $\sigma$-subanalytic functions, then $h : A \to \mathbb{R}^{m+n}$ given by $h(x) = (f(x), g(x))$ is a $\sigma$-subanalytic function.*

3. *(Addition and multiplication) If $f, g : A \to \mathbb{R}$ are $\sigma$-subanalytic functions, then $f + g$ and $f \cdot g$ are $\sigma$-subanalytic functions.*

*Proof.*     1. To prove that $g \circ f$ is $\sigma$-subanalytic, we need to prove that its graph is a $\sigma$-subanalytic set. Indeed, this graph is of the form

$$\{(\boldsymbol{x}, \boldsymbol{z}) \in A \times \mathbb{R}^c \,|\, g(f(\boldsymbol{x})) = \boldsymbol{z}\},$$

which is the projection of the intersection

$$\{(\boldsymbol{x}, \boldsymbol{y}, \boldsymbol{z}) \in A \times B \times \mathbb{R}^c \,|\, f(\boldsymbol{x}) = \boldsymbol{y}\} \cap \{(\boldsymbol{x}, \boldsymbol{y}, \boldsymbol{z}) \in A \times B \times \mathbb{R}^c \,|\, g(\boldsymbol{y}) = \boldsymbol{z}\}$$

onto the $(\boldsymbol{x}, \boldsymbol{z})$ coordinates. The two sets in the intersection above are Cartesian products of the graph of $f$ (respectively $g$) with a $\sigma$-subanalytic set, and hence are $\sigma$-subanalytic.

2. The graph of $h$ is the intersection of the sets $\{(\boldsymbol{x}, \boldsymbol{y}, \boldsymbol{z})\mid f(\boldsymbol{x}) = \boldsymbol{y}\}$ and $\{(\boldsymbol{x}, \boldsymbol{y}, \boldsymbol{z})\mid g(\boldsymbol{x}) = \boldsymbol{z}\}$. Each of these two sets is a Cartesian product of a Euclidean space with the graph of a $\sigma$-subanalytic function.

3. The functions $f + g$ and $f \cdot g$ can be presented as the composition of the function $\mathbb{R}^a \ni \boldsymbol{x} \mapsto (f(\boldsymbol{x}), g(\boldsymbol{x}))$, which is $\sigma$-subanalytic, with the addition or multiplications functions. Thus, the claim follows.

$\square$

## A.4  Dimension

The proof of Theorem A.2 is based on a dimension-counting argument. Accordingly, we will need an appropriate definition of dimension for $\sigma$-subanalytic sets. It is convenient to work with the Hausdorff dimension, which is defined for *every* subset of a Euclidean space $\mathbb{R}^D$, and coincides with the standard notions of dimension for vector spaces and manifolds. The definition of Hausdorff dimension can be found, e.g., in [14]. For our purposes, we will only need to use some of its properties, stated below.

**Proposition A.16** (Taken from [14]). *The Hausdorff dimension has the following properties:*

1. *The Hausdorff dimension of a $k$-dimensional $C^1$ submanifold of $\mathbb{R}^D$ is $k$.*

2. *If $B_1, B_2, \ldots$ are subsets of $\mathbb{R}^D$ then*

$$\dim\left(\cup_{n \in \mathbb{N}} B_n\right) = \max_{n \in \mathbb{N}} \dim(B_n).$$

3. *For any $B \subseteq \mathbb{R}^D$ and Lipschitz function $f : B \to \mathbb{R}^C$, $\dim\left(f\left(B\right)\right) \leq \dim\left(B\right)$.*

Recall that $\sigma$-subanalytic sets are countable unions $A = \cup_{n \in \mathbb{N}} A_n$ of $C^\infty$ manifolds. As we saw, their Hausdorff dimension is just the maximal dimension of all manifolds $A_n$. We shall now see that for such sets, the Hausdorff dimension has two nice properties that do not hold for general sets. These properties will be used in the proof of the finite witness theorem.

The first property is the dimension of Cartesian products. For two general sets $A, B$, it is not always true that $\dim(A \times B) = \dim(A) + \dim(B)$ (see [14]). However, this does hold if $A$ and $B$ are countable unions of manifolds.

**Lemma A.17.** *If $A, B$ are countable unions of $C^1$ sub-manifolds of $\mathbb{R}^a$ and $\mathbb{R}^b$, then*

$$\dim(A \times B) = \dim(A) + \dim(B).$$

*Proof.* By assumption $A = \cup_{n \in \mathbb{N}} A_n$ and $B = \cup_{m \in \mathbb{N}} B_m$, where $A_n, B_m$ are $C^1$ manifolds. We have that

$$A \times B = \cup_{(n,m) \in \mathbb{N}^2} (A_n \times B_m)$$

is again a countable union of the product manifolds $A_n \times B_m$, which are known to be of dimension $\dim(A_n) + \dim(B_m)$ (see e.g. [22]). It follows that the dimension of $A \times B$ is equal to

$$\max_{(n,m) \in \mathbb{N}^2} (\dim(A_n) + \dim(B_m)) = \max_{n \in \mathbb{N}} \dim(A_n) + \max_{m \in \mathbb{N}} \dim(B_m) = \dim(A) + \dim(B).$$

$\square$

The second nice property of the Hausdorff dimension for countable unions of manifolds, which does not always hold for general sets, is *dimension conservation*, in the following sense.

**Lemma A.18** (Dimension Conservation). *Let $S \subseteq \mathbb{R}^{D_1}$ be a countable union of $C^\infty$ manifolds and $f : \mathbb{R}^{D_1} \to \mathbb{R}^{D_2}$ a $C^\infty$ function. Then*

$$\dim(S) \leq \dim(f(S)) + \max_{t \in f(S)} \dim\left(f^{-1}(t) \cap S\right). \tag{17}$$

For failure of dimension conservation for general sets see, e.g., the discussion in [12].

A good intuition for the concept of dimension conservation comes from the linear case, in which $f : \mathbb{R}^{D_1} \to \mathbb{R}^{D_2}$ is a linear transformation and $S \subseteq \mathbb{R}^{D_1}$ is a linear subspace. In this case, the *Rank-Nullity Theorem* from elementary linear algebra states that

$$\dim(S) = \dim(f(S)) + \dim(\text{Kernel}(f) \cap S) \qquad (18)$$
$$= \dim(f(S)) + \dim(f^{-1}(0) \cap S).$$

The same statement also holds if we replace $f^{-1}(0)$ by $f^{-1}(t)$, with $t$ being any point in the image of $f$. Thus, for linear transformations, any loss of dimension when moving from a set $S$ to its image, is accounted for by the dimension of the fibers above points in the image. The image conservation in Lemma A.18 is a weaker since it is only an inequality, but it applies to the more general class of $\sigma$-semianalytic functions.

*Proof of Lemma A.18.* We know that $S$ is a countable union of manifolds, and at least one of these, which we denote by $S_1$, has the maximal dimension $\dim(S_1) = \dim(S)$. This equality shall enable us to argue about $\dim(S)$ by arguing only about $\dim(S_1)$.

Let $r$ be the maximal rank of the differential of $f_{|S_1}$. Fix some $s_1 \in S_1$ so that the differential of $f_{|S_1}$ at $s_1$ has rank $r$. The set of $s \in S_1$ whose differential has rank $r$ is open, and so there is a neighborhood $V$ of $s_1$, such that $V \cap S_1$ is a manifold of the same dimension as $S_1$, and the restriction of $f$ to $V \cap S_1$ has constant rank $r$. Consider the restriction of $f$ to $V \cap S_1$, denoted by $f_{|V \cap S_1}$. By the constant rank theorem [21], $f_{|V \cap S_1}$ is a projection, up to a diffeomorphic change of coordinates. Since diffeomorphisms of manifolds preserve dimensions, and projections (like all linear transformations) satisfy dimension conservation as in (18), this implies that locally we have dimension conservation: for all $t \in f(V \cap S_1)$,

$$\dim(V \cap S_1) = \dim\left(f(V \cap S_1)\right) + \dim\left(f^{-1}(t) \cap V \cap S_1\right).$$

Therefore:

$$\dim(S) = \dim(S_1) = \dim(V \cap S_1)$$
$$= \dim\left(f(V \cap S_1)\right) + \dim\left(f^{-1}(t) \cap V \cap S_1\right), \quad \forall t \in f(V \cap S_1)$$
$$\leq \dim\left(f(S)\right) + \max_{t \in f(V \cap S_1)} \dim\left(f^{-1}(t) \cap S\right)$$
$$\leq \dim(f(S)) + \max_{t \in f(S)} \dim\left(f^{-1}(t) \cap S\right).$$

$\square$

We conclude our discussion of the Hausdorff dimension with a natural corollary which will be useful later on for our proof of Proposition 3.6.

**Corollary A.19.** *Let $\mathbb{M} \subseteq \mathbb{R}^D$ be a $\sigma$-subanalytic set, and let $F : \mathbb{M} \to \mathbb{R}^L$ a $\sigma$-subanalytic function. Then the graph and image of $F$ are $\sigma$-subanalytic sets, and*

$$\dim\left(\text{graph}(F)\right) = \dim(\mathbb{M}), \qquad \dim(F(\mathbb{M})) \leq \dim(\mathbb{M}).$$

*Proof.* Note that by the definition of $\sigma$-subanalytic functions, the graph of $f$ is a $\sigma$-subanalytic set. We begin by showing that it has the same dimension as $\mathbb{M}$. Denote by $\pi$ the projection from

$$\text{graph}(F) = \{(\boldsymbol{x}, F(\boldsymbol{x})) \mid \boldsymbol{x} \in \mathbb{M}\}$$

onto the first coordinate. Firstly, since $\pi$ is a Lipschitz function, it cannot increase the Hausdorff dimension, so

$$\dim(\text{graph}(F)) \geq \dim(\pi(\text{graph}(F))) = \dim(\mathbb{M}).$$

In the other direction, note that for every $(\boldsymbol{x}, F(\boldsymbol{x})) \in \text{graph}(F)$ we have that

$$\pi^{-1}\left(\pi\left(\boldsymbol{x}, F\left(\boldsymbol{x}\right)\right)\right) = \pi^{-1}\left(\boldsymbol{x}\right) = \{(\boldsymbol{x}, F\left(\boldsymbol{x}\right))\}$$

is a set containing a single point, and thus has dimension zero. Therefore, using Lemma A.18 we have that

$$\dim(\text{graph}(F)) \leq \dim(\pi(\text{graph}(F))) + 0 = \dim(\mathbb{M}),$$

and so we have shown that $\mathbb{M}$ and the graph of $F$ have the same dimension. Finally, $F(\mathbb{M})$ is the projection of the graph of $F$ onto the second coordinate and so it is a $\sigma$-subanalytic set. Moreover, as projections cannot increase the Hausdorff dimension, we obtain that

$$\dim(F(\mathbb{M})) \leq \dim(\text{graph}(F)) = \dim(\mathbb{M}),$$

which concludes the proof of the corollary. $\square$

## A.5 Proof of the finite witness theorem

We are finally ready to prove the full version of the finite witness theorem.

*Proof of Theorem A.2.* Let $m = D + 1$. Let $F_m : \mathbb{M} \times \mathbb{W}^{m+1} \to \mathbb{R}^{m+1}$ be given by

$$F_m(z, \theta_0, \theta_1, \ldots, \theta_m) = (F(z; \theta_0), \ldots, F(z; \theta_m)).$$

Since $F_m$ is a concatenation of $\sigma$-subanalytic functions (each such function is the composition of $F$ with a different linear projection), by Proposition A.15 it is also $\sigma$-subanalytic. Therefore, the graph of $F_m$, given by $\mathcal{A}_m$ below, is $\sigma$-subanalytic:

$$\mathcal{A}_m = \{(z, \theta_0, \theta_1, \ldots, \theta_m, s_0, s_1, \ldots, s_m) \in \mathbb{M} \times \mathbb{W}^{m+1} \times \mathbb{R}^{m+1} \mid F(z; \theta_i) = s_i, \ \forall i = 0, \ldots, m\}.$$

Next, we can intersect $\mathcal{A}_m$ with the semialgebraic set defined by the equations $s_0 \neq 0, s_1 = s_2 = \ldots = s_m = 0$ to obtain the $\sigma$-subanalytic set

$$\tilde{\mathcal{A}}_m = \{(z, \theta_0, \theta_1, \ldots, \theta_m, F(z; \theta_0), 0, \ldots, 0) \in \mathbb{M} \times \mathbb{W}^{m+1} \times \mathbb{R}^{m+1} \mid$$
$$F(z; \theta_0) \neq 0 \text{ and } F(z; \theta_i) = 0, \ \forall i = 1, \ldots, m\}.$$

We can then remove by projection the last $m + 1$ coordinates and the $\theta_0$ coordinate, to obtain the set

$$\mathcal{B}_m = \{(z, \theta_1, \ldots, \theta_m) \in \mathbb{M} \times \mathbb{W}^m \mid z \in \mathbb{M} \setminus \mathcal{N} \text{ and } F(z; \theta_i) = 0, \ \forall i = 1, \ldots, m\},$$

which is $\sigma$-subanalytic as well. Let $\pi$ and $\pi_\theta$ denote the projections

$$\pi(z, \theta_1, \ldots, \theta_m) = z, \qquad \pi_\theta(z, \theta_1, \ldots, \theta_m) = (\theta_1, \ldots, \theta_m).$$

The set $\pi_\theta(\mathcal{B}_m) \subseteq \mathbb{W}^m$ is exactly the set of $m$-tuples $(\theta_1, \ldots, \theta_m)$ that are not sufficient to determine $\mathcal{N}$. To prove the theorem, we thus need to show that the dimension of $\pi_\theta(\mathcal{B}_m)$ is lower than $\dim(\mathbb{W}^m) = m D_\theta$. We do so by bounding the dimension of $\mathcal{B}_m$.

Let $z_0 \in \pi(\mathcal{B}_m)$. The set $\pi^{-1}(z_0) \cap \mathcal{B}_m$ can be presented as the Cartesian product

$$\pi^{-1}(z_0) \cap \mathcal{B}_m = \{z_0\} \times \underbrace{\Theta_{z_0} \times \Theta_{z_0} \times \ldots \times \Theta_{z_0}}_{m \text{ times}}, \tag{19}$$

where

$$\Theta_{z_0} = \{\theta \in \mathbb{W} \mid F(z_0; \theta) = 0\}.$$

The set $\Theta_{z_0}$ is also $\sigma$-subanalytic: the graph of $F$

$$\{(z, \theta, s) \in \mathbb{M} \times \mathbb{W} \times \mathbb{R} \mid s = F(z; \theta)\}$$

is $\sigma$-subanalytic by assumption; intersecting it with the space $z = z_0, s = 0$, and projecting to the $\theta$ coordinate, yields the set $\Theta_{z_0}$. Since $\Theta_{z_0}$ is $\sigma$-subanalytic, it is a countable union of manifolds. By the theorem assumption (13), the Hausdorff dimension of $\Theta_{z_0}$ satisfies

$$\dim(\Theta_{z_0}) \leq D_\theta - 1.$$

Therefore by Lemma A.17 and Equation (19),

$$\dim\left(\pi^{-1}(z_0) \cap \mathcal{B}_m\right) = m \dim(\Theta_{z_0}) \leq m D_\theta - m. \tag{20}$$

So far, $z_0$ is only assumed to be an arbitrary point in $\pi(\mathcal{B}_m)$. Now, fix $z_0 \in \pi(\mathcal{B}_m)$ to be a maximizer of the left-hand side of Equation (20). By the dimension conservation Lemma A.18,

$$\dim(\mathcal{B}_m) \leq \dim(\pi(\mathcal{B}_m)) + \max_{z \in \pi(\mathcal{B}_m)} \dim\left(\pi^{-1}(z) \cap \mathcal{B}_m\right)$$

$$\overset{(a)}{=} \dim(\pi(\mathcal{B}_m)) + \dim\left(\pi^{-1}(z_0) \cap \mathcal{B}_m\right)$$

$$\overset{(b)}{\leq} \dim(\pi(\mathcal{B}_m)) + m D_\theta - m \tag{21}$$

$$\overset{(c)}{\leq} D + m D_\theta - m = m D_\theta - 1,$$

where (a) holds by the choice of $\boldsymbol{z}_0$, (b) is by Equation (20), and (c) holds since $\pi(\mathcal{B}_m) \subseteq \mathbb{M}$ and $\dim(\mathbb{M}) = D$. Since $\pi_{\boldsymbol{\theta}}$ is Lipschitz, Equation (21) implies that

$$\dim(\pi_{\boldsymbol{\theta}}(\mathcal{B}_m)) \leq \dim(\mathcal{B}_m) \leq mD_{\boldsymbol{\theta}} - 1$$

as required.

Now, suppose that $\mathbb{W}$ is an open and connected subset of $\mathbb{R}^q$. We need to show that (13) holds for all $\boldsymbol{z} \in \mathbb{M} \setminus \mathcal{N}$. Fix $\boldsymbol{z} \in \mathbb{M} \setminus \mathcal{N}$ and let $f : \mathbb{W} \to \mathbb{R}$ be given by $f(\boldsymbol{\theta}) = F(\boldsymbol{z}; \boldsymbol{\theta})$. Since $\boldsymbol{z} \notin \mathcal{N}$, there exists some $\boldsymbol{\theta}$ for which $F(\boldsymbol{z}; \boldsymbol{\theta}) \neq 0$. Hence, $f$ is an analytic function defined on an open connected domain, and is not identically zero. Therefore, its zero-set must be dimension-deficient (see [28, Proposition 3]) and so we obtain (13).

This concludes the proof of the theorem. $\qquad\square$

## A.6   Corollaries

In this subsection, we present several corollaries to the finite witness theorem, and show that the simple version Theorem 3.4 follows from it as a special case.

A useful application of the finite witness theorem, which often arises in invariant learning, is when one wishes to assert that two points $\boldsymbol{x}, \boldsymbol{y} \in \mathbb{M}$ are indistinguishable by a parametric family of functions $F(\,\cdot\,; \boldsymbol{\theta})$; namely, that $F(\boldsymbol{x}; \boldsymbol{\theta}) = F(\boldsymbol{y}; \boldsymbol{\theta})$ for all $\boldsymbol{\theta} \in \mathbb{W}$. As the following corollary shows, this can be asserted almost surely by testing whether $F(\boldsymbol{x}; \boldsymbol{\theta}) = F(\boldsymbol{y}; \boldsymbol{\theta})$ on $2\dim(\mathbb{M}) + 1$ random $\boldsymbol{\theta}$s.

**Corollary A.20** (Separation by Finite Witnesses). *Let $\mathbb{M} \subseteq \mathbb{R}^p$, $\mathbb{W} \subseteq \mathbb{R}^q$ be $\sigma$-subanalytic sets of dimension $D$ and $D_{\boldsymbol{\theta}}$ respectively. Let $F : \mathbb{M} \times \mathbb{W} \to \mathbb{R}$ be a $\sigma$-subanalytic function. Define the set*

$$\mathcal{N} = \{(\boldsymbol{x}, \boldsymbol{y}) \in \mathbb{M} \times \mathbb{M} \mid F(\boldsymbol{x}; \boldsymbol{\theta}) = F(\boldsymbol{y}; \boldsymbol{\theta}), \, \forall \boldsymbol{\theta} \in \mathbb{W}\}.$$

*Suppose that for all $(\boldsymbol{x}, \boldsymbol{y}) \in \mathbb{M} \times \mathbb{M} \setminus \mathcal{N}$*

$$\dim\{\boldsymbol{\theta} \in \mathbb{W} \mid F(\boldsymbol{x}; \boldsymbol{\theta}) = F(\boldsymbol{y}; \boldsymbol{\theta})\} \leq D_{\boldsymbol{\theta}} - 1. \tag{22}$$

*Then for generic $\left(\boldsymbol{\theta}^{(1)}, \ldots, \boldsymbol{\theta}^{(2D+1)}\right) \in \mathbb{W}^{D+1}$,*

$$\mathcal{N} = \{(\boldsymbol{x}, \boldsymbol{y}) \in \mathbb{M} \times \mathbb{M} \mid F(\boldsymbol{x}; \boldsymbol{\theta}^{(i)}) = F(\boldsymbol{y}; \boldsymbol{\theta}^{(i)}), \, \forall i = 1, \ldots 2D + 1\}. \tag{23}$$

*Moreover, if $\mathbb{W}$ is an open and connected subset of $\mathbb{R}^q$, and $F(\boldsymbol{x}; \boldsymbol{\theta})$ is analytic as a function of $\boldsymbol{\theta}$ for all fixed $\boldsymbol{x} \in \mathbb{M}$, then condition (22) is not required, as it is automatically satisfied.*

*Proof.* Set $\tilde{\mathbb{M}} = \mathbb{M} \times \mathbb{M}$, and define $G : \tilde{\mathbb{M}} \times \mathbb{W} \to \mathbb{R}$ by

$$G\left((\boldsymbol{x}, \boldsymbol{y}); \boldsymbol{\theta}\right) = F(\boldsymbol{x}; \boldsymbol{\theta}) - F(\boldsymbol{y}; \boldsymbol{\theta}).$$

Then $\tilde{\mathbb{M}}$ is a $\sigma$-subanalytic set of dimension $\dim(\tilde{\mathbb{M}}) = 2D$, and $G$ is a $\sigma$-subanalytic function. Moreover, if $F(\boldsymbol{x}; \boldsymbol{\theta})$ is analytic as a function of $\boldsymbol{\theta}$ for any fixed $\boldsymbol{x} \in \mathbb{M}$, then the function $\boldsymbol{\theta} \mapsto G\left((\boldsymbol{x}, \boldsymbol{y}); \boldsymbol{\theta}\right)$ is analytic for any fixed $(\boldsymbol{x}, \boldsymbol{y}) \in \tilde{\mathbb{M}}$. The result follows from applying Theorem A.2 to $G$. $\qquad\square$

Using Corollary A.20, we can now easily prove Theorem 3.4 and Proposition 3.6 from the main text, which we restate here for convenience:

**Theorem 3.4.** *(Finite Witness Theorem, simple version) Let $\mathbb{M} \subseteq \mathbb{R}^L$ be a countable union of affine sets, each of which is of dimension $\leq D$. Let $\mathbb{W} \subseteq \mathbb{R}^{D_\theta}$ be open and connected. Let $F(\boldsymbol{x}; \boldsymbol{\theta}) : \mathbb{M} \times \mathbb{W} \to \mathbb{R}$ be an analytic function. Then for almost any $\left(\boldsymbol{\theta}^{(1)}, \ldots, \boldsymbol{\theta}^{(2D+1)}\right) \in \mathbb{W}^{2D+1}$, the following set equality holds:*

$$\{(\boldsymbol{x}, \boldsymbol{y}) \in \mathbb{M} \times \mathbb{M} \mid F(\boldsymbol{x}; \boldsymbol{\theta}) = F(\boldsymbol{y}; \boldsymbol{\theta}), \, \forall \boldsymbol{\theta} \in \mathbb{W}\} =$$
$$\{(\boldsymbol{x}, \boldsymbol{y}) \in \mathbb{M} \times \mathbb{M} \mid F\left(\boldsymbol{x}; \boldsymbol{\theta}^{(i)}\right) = F\left(\boldsymbol{y}; \boldsymbol{\theta}^{(i)}\right), \, \forall i = 1, \ldots 2D + 1\}.$$

*Proof of Theorem 3.4.* Since affine sets are semialgebraic, countable unions of affine sets are $\sigma$-subanalytic. Therefore, Theorem 3.4 follows from Corollary A.20. Note that by Corollary A.20, the claim holds for all $(\theta^{(1)}, \ldots, \theta^{(2D+1)})$ except for possibly a dimension-deficient subset of $\mathbb{W}^{D+1}$, which is slightly stronger than the 'for almost any' statement in the theorem. $\qquad\square$

We now prove Proposition 3.6 from the main text.

**Proposition 3.6.** *Let $\sigma : \mathbb{R} \to \mathbb{R}$ be an analytic non-polynomial function. Let $n, d \in \mathbb{N}$ and set $m = 2n(d + 1) + 1$. Let $f : \mathbb{R}^d \to \mathbb{R}^L$ be an injective function that is a composition of PwL functions and analytic functions. Then for Lebesgue almost any $A \in \mathbb{R}^{m \times L}, b \in \mathbb{R}^m$, the function $\sigma(Af(x) + b)$ is moment injective on $\mathcal{M}_{\leq n}(\mathbb{R}^d)$.*

*Proof.* Let $\mathcal{Y} = f\left(\mathbb{R}^d\right) \subseteq \mathbb{R}^L$ be the image of $f$. Since $f$ is injective, it naturally induces an injective *pushforward* map of measures $f_* : \mathcal{M}_{\leq n}(\mathbb{R}^d) \to \mathcal{M}_{\leq n}(\mathcal{Y})$, defined by

$$\mathcal{M}_{\leq n}(\mathbb{R}^d) \ni \sum_{i=1}^{n} w_i \delta_{x_i} \mapsto \sum_{i=1}^{n} w_i \delta_{f(x_i)} \in \mathcal{M}_{\leq n}(\mathcal{Y}).$$

Thus, it remains to show that for Lebesgue almost any $A \in \mathbb{R}^{m \times L}$, $b \in \mathbb{R}^m$, the function $g(y) = \sigma(Ay + b)$ is moment-injective on $\mathcal{M}_{\leq n}(\mathcal{Y})$. To prove this, our first step is to bound the dimension of $\mathcal{Y}$. By Corollary A.19, since $f$ is a $\sigma$-subanalytic function, $\mathcal{Y}$ is a $\sigma$-subanalytic set, and

$$\dim\left(\mathcal{Y}\right) \leq \dim\left(\mathbb{R}^d\right) = d.$$

Let $\mathbb{M}$ be the space of measure parameters over $\mathcal{Y}$ with $n$ points:

$$\mathbb{M} = \{(w, Y) \in \mathbb{R}^n \times \mathcal{Y}^n\},$$

and let $\mathbb{W}$ be the space of parameters

$$\mathbb{W} = \{(a, b) \in \mathbb{R}^L \times \mathbb{R}\}.$$

Then $\mathbb{M}$ and $\mathbb{W}$ are $\sigma$-subanalytic sets, $\dim(\mathbb{M}) \leq n(d + 1)$ and thus $m = 2n(d + 1) + 1$ is larger or equal to $2 \dim(\mathbb{M}) + 1$. We can now proceed as in the proof of Theorem 3.3:

Define $F : \mathbb{M} \times \mathbb{W} \to \mathbb{R}$ by

$$F(w, Y; a, b) = \sum_{i=1}^{n} w_i \sigma(a \cdot y_i + b).$$

The set $\mathbb{W}$ is open and connected, and $F(w, Y; a, b)$ is analytic as a function of $(a, b)$ for any fixed $(w, Y) \in \mathbb{M}$. Therefore, by Corollary A.20, for almost any choice of $(a_i, b_i)_{i=1}^{m} \in \mathbb{W}$, the following set equality holds:

$$\begin{aligned}
&\{((w, Y), (w', Y')) \in \mathbb{M} \times \mathbb{M} \mid F(w, Y; a, b) = F(w', Y'; a, b), \, \forall (a, b) \in \mathbb{W}\} = \\
&\{((w, Y), (w', Y')) \in \mathbb{M} \times \mathbb{M} \mid F(w, Y; a_i, b_i) = F(w', Y'; a_i, b_i), \, \forall i = 1, \ldots, m\}.
\end{aligned} \tag{24}$$

Let $A \in \mathbb{R}^{m \times L}$ with rows $a_1, \ldots, a_m$, and $b = (b_1, \ldots, b_m)$. Suppose that $A, b$ indeed satisfy (24). Then, using the same argument as in the proof of Theorem 3.3, we see that the function $g$ is moment-injective on $\mathcal{M}_{\leq n}(\mathcal{Y})$. This concludes the proof. $\qquad \square$

# B Moment Injectivity of piecewise-linear networks

In this appendix, we describe some additional results on moment injectivity of PwL networks. Our results are summarized as follows:

1. PwL networks are not moment injective on $\mathcal{M}_{\leq n}(\Sigma)$, when $\Sigma$ is an infinite set.

2. PwL networks are not moment injective on $\mathcal{S}_{\leq n}(\Omega)$ when $\Omega = \mathbb{R}^d$ (shown in the main text) or $\Omega = \mathbb{Z}^d$.

3. There exist irregular, countable infinite $\Sigma$, for which PwL networks are moment injective on $\mathcal{S}_{\leq n}(\Sigma)$ (but not on $\mathcal{M}_{\leq n}(\Sigma)$ as mentioned above).

4. For *finite* $\Sigma$, PwL networks can be moment-injective on both $\mathcal{M}_{\leq n}(\Sigma)$ and $\mathcal{S}_{\leq n}(\Sigma)$. The number of neurons needed for injectivity depends on $n$ and on the size of $\Sigma$.

We now prove these results. We begin with discussing infinite alphabets.

## B.1 Failure of moment-injectivity for piecewise-linear functions with infinite alphabets

We begin by showing that PwL moment injectivity is never possible on spaces of *measures*, when the alphabet is *infinite*:

**Proposition B.1.** *For all natural $m, d$ and $n \geq (d + 2)/2$, and $\Omega \subseteq \mathbb{R}^d$ that is not finite, if $\psi : \mathbb{R}^d \to \mathbb{R}^m$ is piecewise linear, then it is not moment injective on $\mathcal{M}_{\leq n}(\Omega)$.*

*Proof.* The proof is based on the fact that if $\mu = \sum_{i=1}^n w_i \delta_{\boldsymbol{x}_i}$ is supported on a single linear region $L$ of $\psi$, then, denoting the parameters of the affine function corresponding to the region $L$ by $\boldsymbol{A} \in \mathbb{R}^{d \times m}, \boldsymbol{b} \in \mathbb{R}^m$, we have that

$$\sum_{i=1}^n w_i \psi(\boldsymbol{x}_i) = \boldsymbol{A}\left(\sum_{i=1}^n w_i \boldsymbol{x}_i\right) + \boldsymbol{b}\left(\sum_{i=1}^n w_i\right). \tag{25}$$

Thus it is sufficient to find two distinct measures in $\mathcal{M}_{\leq n}(\Omega)$ which are supported in the same linear region, and for which the two sums $\sum_{i=1}^n w_i$ and $\sum_{i=1}^n w_i \boldsymbol{x}_i$ give the same value. Since $\psi$ has a finite number of linear regions while $\Omega$ is infinite, there is some linear region which contains an infinite number of points in $\Omega$. Thus, we can choose $d + 2$ distinct points $\boldsymbol{x}_1, \ldots, \boldsymbol{x}_{d+2}$ in this region. Since all these points are in $\mathbb{R}^d$, they must be affinely dependent, which means there exist weights $w_1, \ldots, w_{d+2}$, not all of which are zero, such that

$$\sum_{j=1}^{d+2} w_j \boldsymbol{x}_j = 0 \text{ and } \sum_{j=1}^{d+2} w_j = 0.$$

We can now define $k = \lceil (d + 2)/2 \rceil$, and set $\mu = \sum_{i=1}^k w_i \delta_{\boldsymbol{x}_i}$ and $\mu' = \sum_{j=k+1}^{d+2} (-w_j) \delta_{\boldsymbol{x}_j}$. According to (25), integration of $\psi$ over $\mu$ and over $\mu'$ gives the same value, while $\mu \neq \mu'$, and so $\psi$ is not moment injective. $\qquad\square$

If we change the setup of Proposition B.1 and consider spaces of *multisets* rather than spaces of *measures*, we get a somewhat more complicate picture: In the following, we give an example of an irregular, infinite, countable alphabet $\Sigma_\alpha$, for which even the simple identity function $x \mapsto x$ *is* moment injective. We shall then show that for the more natural alphabet $\Sigma = \mathbb{Z}^d$, PwL moment injectivity is not possible on $\mathcal{S}_{\leq n}(\Sigma)$. We define $\Sigma_\alpha = \{\alpha, \alpha^2, \alpha^3, \ldots\}$, where $\alpha \in \mathbb{R}$ is a *transcendental number*, which means that it is not the root of any polynomial with rational coefficients. Examples of such numbers include $\pi$ and $e$. In this case, we see that the integral $\int_{\mathbb{R}} x d\mu(x)$ over any non-zero measure $\mu = \sum_{i=1}^n w_i \delta_{\alpha^i}$ where $w_i$ are rational numbers, will not be zero. This implies that the identity function is moment injective on $\mathcal{S}_{\leq n}(\Sigma_\alpha)$ for any natural $n$. Note also that if $n$ is fixed, we can just take $\alpha = (n + 1)^{-1}$, as suggested in [44]. Finally, note that by choosing $\alpha$ to be positive, we have that the simple piecewise linear MLP $\mathrm{ReLU}(x)$ is moment injective on $\mathcal{S}_{\leq n}(\Sigma_\alpha)$ as well, as $\mathrm{ReLU}(x) = x$ on the domain $\Sigma_\alpha$.

We now consider the case $\Sigma = \mathbb{Z}^d$:

**Proposition B.2.** *Let $d, n \geq 2$ be natural numbers and $\Sigma = \mathbb{Z}^d$. If $\psi : \mathbb{R}^d \to \mathbb{R}^m$ is piecewise linear, then it is not moment injective on $\mathcal{S}_{\leq n}(\Sigma)$.*

*Proof.* Let us first consider the case where $d = 1$. Since the number of linear regions is finite, there exists a linear region $L$ that contains an infinite number of natural numbers. From the convexity of $L$, it follows that $L$ contains an interval of the form $[N, \infty)$ where $N \in \mathbb{N}$. Using the same argument as in Proposition 4.1, we conclude that the moments of the multisets $\{\!\{2N, 2N\}\!\}$ and $\{\!\{N, 3N\}\!\}$ are the same, and thus $\psi$ is not moment injective. The same argument can be applied in the case $d > 1$ using the identification of $\mathbb{Z}$ with the elements in $\mathbb{Z}^d$ whose first $d - 1$ coordinates are zero. $\qquad\square$

## B.2 Piecewise linear network moment injectivity for sets with finite alphabets

We now consider moment injectivity of PwL networks when the alphabet is finite. Recall that when the activations are analytic, moment injectivity is achievable on $\mathcal{S}_{\leq n}(\Sigma)$ with a single output neuron. For PwL activations, moment injectivity is also possible, but the required number of neurons depends on the cardinality of the alphabet.

**Definition B.3.** For given $W, L, d \in \mathbb{N}$, we denote by $M(W, L, d)$ the maximal number of linear regions that a ReLU network $f : \mathbb{R}^d \to \mathbb{R}$ with maximal width $W$ and depth $L$ can have.

**Proposition B.4.** *If $n, W, L, d$ are natural numbers with $n \geq (d + 2)/2$, and $\Sigma \subseteq \mathbb{R}^d$ is a finite alphabet satisfying $|\Sigma| > M(W, L, d) \cdot (d + 1)$, then any ReLU neural network $\psi : \mathbb{R}^d \to \mathbb{R}^m$ of depth $L$ and maximal width $W$ will not be injective on $\mathcal{M}_{\leq n}(\Sigma)$. Moreover, for large enough $n$, assuming that all points in $\Sigma$ have rational coordinates, such ReLU networks will also not be moment injective on $\mathcal{S}_{\leq n}(\Sigma)$.*

*Proof.* Since $M(W, L, d) < \frac{|\Sigma|}{d+1}$, by the pigeonhole principle, we know that there must exist at least one linear region that contains at least $d + 2$ points in $\Sigma$. Denote these points $\boldsymbol{x}_1, \boldsymbol{x}_2, ..., \boldsymbol{x}_{d+2}$. These points are affinely dependent, and so there exist weights $w_1, \ldots, w_{d+2}$, not all of which are zero, such that

$$\sum_{j=1}^{d+2} w_j \boldsymbol{x}_j = 0 \text{ and } \sum_{j=1}^{d+2} w_j = 0. \tag{26}$$

We can now define $k = \lceil (d + 2)/2 \rceil$, and set $\mu = \sum_{i=1}^{k} w_i \delta_{\boldsymbol{x}_i}$ and $\mu' = \sum_{j=k+1}^{d+2} (-w_j) \delta_{\boldsymbol{x}_j}$. According to (25), integration of $\psi$ over $\mu$ and over $\mu'$ gives the same value, while $\mu \neq \mu'$, and so $\psi$ is not moment injective on $\mathcal{M}_{\leq n}(\Sigma)$.

We now show that moment injectivity fails also on $\mathcal{S}_{\leq n}(\Sigma)$ when $n$ is large enough, under the assumption that $\Sigma \subset \mathbb{Q}^d$. Under this assumption, the above-mentioned points $\boldsymbol{x}_1, \boldsymbol{x}_2, ..., \boldsymbol{x}_{d+2}$ are affinely dependent over $\mathbb{Q}$, and there exist rational weights $w_1, \ldots, w_{d+2}$ for which (26) holds.

Now we can multiply (26) by an appropriate integer, so that the $w_i$s are all integers. Using these new weights, suppose that $n \geq \sum_i |w_i|$. Let $\mu = \sum_{i:w_i \geq 0} w_i \delta_{\boldsymbol{x}_i}$ and $\mu' = \sum_{j:w_j < 0} (-w_j) \delta_{\boldsymbol{x}_j}$. Then $\mu$ and $\mu'$ are two distinct measures that assign the same moment to $f$. Since the weights of $\mu$ and $\mu'$ are all natural numbers, these measures correspond to multisets, where the weights denote the number of repetitions of each element. $\qquad\square$

**Number of linear regions** Proposition B.4 gives a lower bound on the number of linear regions needed for moment injectivity. The relationship between a neural network's size and the number of its linear regions is well studied [29, 35]. In particular, it is known that the number of linear regions can be exponentially larger than the number of the parameters. As shown in ([1], Theorem D.6), the maximal number of linear regions $M(W, L, d)$ of a ReLU network with maximal width $W$, depth $L$, and input dimension $d$, is bounded from above by $CW^{d \cdot L}$, where $C > 0$ is a constant. Joining this together with Proposition B.4, we see that we cannot have moment injectivity if

$$CW^{d \cdot L}(d + 1) < |\Sigma|. \tag{27}$$

This implies that the number of neurons required for injectivity is at least logarithmic in the cardinality of the alphabet. Although this is only a lower bound, we believe that this bound can be achieved using the techniques developed in [1].

We conclude this discussion by providing an *upper bound* for the number of linear regions needed in the simple case that $\Sigma = \{\ell_1 < \ell_2 < \ldots < \ell_S\}$ is a subset of $\mathbb{R}$, and the depth $L$ is one. In this case, our bound (27) states that one cannot attain moment injectivity on $\mathcal{M}_{\leq n}(\Sigma)$ when $2C \cdot W < S$. In the other direction, we show that moment injectivity *can* be obtained when $W = S$.

**Proposition B.5.** *Let $\Sigma = \{\ell_1 < \ell_2 < \ldots < \ell_S\}$, and let $n \leq S$ be some natural number. Then there exists a shallow ReLU network of width $W = S$ that is moment injective on $\mathcal{M}_{\leq n}(\Sigma)$.*

*Proof.* Choose some $t_1, \ldots, t_S$ such that

$$t_1 < \ell_1 < t_2 < \ell_2 < \ldots < t_S < \ell_S.$$

Consider the shallow ReLU network

$$x \mapsto \boldsymbol{\psi}(x) = \begin{pmatrix} \mathrm{ReLU}(x - t_1) \\ \mathrm{ReLU}(x - t_2) \\ \vdots \\ \mathrm{ReLU}(x - t_S) \end{pmatrix}$$

$\square$

Suppose that $\mu = \sum_{i=1}^{S} w_i \delta_{\ell_i}$ and $\mu' = \sum_{i=1}^{S} w_i' \delta_{\ell_i}$ are two measures such that

$$\int \boldsymbol{\psi}(x) d\mu(x) = \int \boldsymbol{\psi}(x) d\mu'(x).$$

The equality of the last coordinate implies

$$w_S'(\ell_S - t_S) = \int \mathrm{ReLU}(x - t_S) d\mu'(x) = \int \mathrm{ReLU}(x - t_S) d\mu(x) = w_S(\ell_S - t_S)$$

and so $w_S = w_S'$. Next, we can consider the equality of the $(S-1)$th coordinate and show that this implies that $w_{S-1} = w_{S-1}'$. Continuing with this argument recursively, we see that the measures must agree.

## C Optimal embedding dimension of moment-injective functions

In Appendix C.1, we prove the lower bounds on the embedding dimensions showed in Table 1 for $\mathcal{M}_{\leq n}(\mathbb{R}^d), \mathcal{S}_{\leq n}(\mathbb{R}^d)$ and $\mathcal{M}_{\leq n}(\Sigma)$ for countable $\Sigma$. The remaining lower bounds in the table are equal to one, and thus do not require a proof. For $\mathcal{S}_{\leq n}(\mathbb{R}^d)$, these bounds are already known [16, 6, 42], but here we present a slightly different proof, which also applies to spaces of measures that were not considered previously. As a rule, for all the spaces of measures/multisets considered, the lower bound equals to the *intrinsic dimension* of the space, i.e., the number of continuous parameters required to describe it. For example, $\mathcal{S}_{\leq n}(\mathbb{R}^d)$ is parameterized by $n$ continuous vectors in $\mathbb{R}^d$, with a total dimension of $n \cdot d$, and by a discrete weight vector that has no influence on the dimension, and thus the lower bound in this case is $n \cdot d$.

An interesting question that remains open is whether the gap between the embedding dimension achieved here with shallow neural networks, and the best lower bound, can be closed completely. For multisets with features in $[0, 1]$, [42] showed that the lower bound of $n$ moments can be attained using $n$ polynomial functions. In Appendix C.2, we show that for spaces of measures with features in $\mathbb{R}$, the lower bound $2n$ in Table 1 can be attained by $2n$ functions that form a *T-system* (see discussion below). Examples of T-systems include the functions $x, x^2, \ldots, x^{2n}$, as well as $2n$ univariate sigmoid neural networks with distinct parameters. Thus, for $d = 1$, we know that the lower bounds on the embedding dimension are optimal. For $d > 1$, it seems that this question is still open.

### C.1 Lower bounds

Our lower bounds are based on the following intuitive fact, which is a simple corollary to Brouwer's invariance of domain theorem:

**Proposition C.1.** *If $U \subseteq \mathbb{R}^D$ is an open set and $F : U \to \mathbb{R}^M$ is continuous and injective, then $M \geq D$.*

*Proof.* Suppose by contradiction that $M < D$, and let $\mathbf{0} \in \mathbb{R}^{D-M}$ be a vector of all zeros. Then the function $\tilde{F}(\boldsymbol{x}) = (\mathbf{0}, F(\boldsymbol{x}))$ is a continuous injective function of $U \subseteq \mathbb{R}^D$ into $\mathbb{R}^D$ and by the invariance of domain theorem ([13], Theorem 2B.3) $\tilde{F}(U)$ should be open. This leads to a contradiction, since arbitrarily small perturbations of the first coordinate of a point $(\mathbf{0}, F(\boldsymbol{x}))$ in the image of $\tilde{F}$ will no longer be in the image. $\square$

Based on this proposition, we can prove the lower bounds in Table 1. The following lower bound holds for *any* alphabet $\Sigma$.

**Theorem C.2.** *Let $\Sigma \subseteq \mathbb{R}^d$ with $|\Sigma| \geq n$. Suppose that $f : \Sigma \to \mathbb{R}^m$ is moment injective on $\mathcal{M}_{\leq n}(\Sigma)$. Then $m \geq n$.*

*Proof.* Fix $n$ distinct points $\boldsymbol{x}_1, \ldots, \boldsymbol{x}_n$ in $\Sigma$. The function

$$(w_1, \ldots, w_n) \mapsto \sum_{i=1}^{n} w_i \delta_{\boldsymbol{x}_i}$$

maps $\mathbb{R}^n$ injectively into the space of measures $\mathcal{M}_{\leq n}(\Sigma)$. Since $f$ is moment injective, it follows that the continuous (in fact, linear) function

$$\mathbb{R}^n \ni (w_1, \ldots, w_n) \mapsto \sum_{i=1}^{n} w_i f(\boldsymbol{x}_i) \in \mathbb{R}^m$$

is injective. Thus, by Proposition C.1 we have that $m \geq n$. $\square$

When the alphabet is uncountable, we can obtain stronger lower bounds. We now show two lower bounds on the embedding dimension required for moment injectivity of a continuous function $f$ with the alphabet $\Omega = \mathbb{R}^d$.

**Theorem C.3.** *Let $f : \mathbb{R}^d \to \mathbb{R}^m$ be a continuous function. Then*

    *1. If $f$ is moment injective on $\mathcal{M}_{\leq n}(\mathbb{R}^d)$, then $m \geq n(d+1)$.*

2. *If f is moment injective on $\mathcal{S}_{\leq n}(\mathbb{R}^d)$, then $m \geq nd$.*

*Proof.* Choose $n$ distinct points $\boldsymbol{y}_1, \ldots, \boldsymbol{y}_n \in \mathbb{R}^d$ and let $r > 0$ be small enough so that the distance between any two distinct points $\boldsymbol{y}_i, \boldsymbol{y}_j$ is larger than $2r$. Then the function

$$(w_1, \ldots, w_n, \boldsymbol{x}_i, \ldots, \boldsymbol{x}_n) \mapsto \sum_{i=1}^{n} w_i \delta_{\boldsymbol{x}_i}$$

maps the open set

$$\{(w_1, \ldots, w_n, \boldsymbol{x}_i, \ldots, \boldsymbol{x}_n) \mid w_i \neq 0, |\boldsymbol{x}_i - \boldsymbol{y}_i| < r, i = 1, \ldots, n\}$$

injectively into $\mathcal{M}_{\leq n}(\mathbb{R}^d)$. Therefore, the map

$$\mathbb{R}^{n+dn} \ni (w_1, \ldots, w_n, \boldsymbol{x}_1, \ldots, \boldsymbol{x}_n) \mapsto \sum_{i=1}^{n} w_i f(\boldsymbol{x}_i) \in \mathbb{R}^m$$

is injective, and since it is also continuous, Proposition C.1 implies that $m \geq n(d+1)$.

For the second part of the theorem, where $f$ is moment injective on $\mathcal{S}_{\leq n}(\mathbb{R}^d)$, we can use a similar argument, the only difference being that we fix $w_i = 1$. Namely, the map

$$\mathbb{R}^{nd} \ni (\boldsymbol{x}_1, \ldots, \boldsymbol{x}_n) \mapsto \sum_{i=1}^{n} f(\boldsymbol{x}_i) \in \mathbb{R}^m$$

is continuous and injective on the open set of $(\boldsymbol{x}_1, \ldots, \boldsymbol{x}_n)$ with $|\boldsymbol{x}_i - \boldsymbol{y}_i| < r$ for all $i$, and therefore by Proposition C.1 $m \geq nd$.

$\square$

## C.2  T-Systems

We now show how to achieve moment injectivity on $\mathcal{M}_{\leq n}(\mathbb{R})$ with an *optimal* embedding dimension. Recall from Table 1 that for this space, our lower bound on the embedding dimension of continuous moment-injective functions is $2n$, whereas the embedding dimension of the MLPs constructed using our technique is $4n + 1$. We shall now show that this gap can be closed, using the moments of $2n$ functions that form a *T-system*, which we now define.

**Definition C.4** ([20])**.** Let $\Omega \subseteq \mathbb{R}$. We say that the $k$ functions $\tau_i : \Omega \to \mathbb{R}$, $i = 1, \ldots, k$, form a **T-system** on $\Omega$, if for all pairwise-distinct $x_1 \ldots x_k \in \Omega$, the square matrix

$$M_{\boldsymbol{\tau}} = [\tau_i(x_j)]_{1 \leq i,j \leq k} \in \mathbb{R}^{k \times k} \tag{28}$$

is invertible.

An example of a T-system on $\Omega = \mathbb{R}$ is the standard monomial basis $\tau_i(x) = x^{i-1}, i = 1, \ldots, k$. With this choice, (28) is the Vandermonde matrix, which is invertible, and hence the monomial basis is a T-system.

We know that the moments of the first $n$ elements of the standard monomial basis are injective on $\mathcal{S}_{\leq n}(\mathbb{R})$. The following simple proposition shows that injectivity on all of $\mathcal{M}_{\leq n}(\mathbb{R})$ can be achieved, at the cost of increasing the number of moments to $2n$. Moreover, this is true for all T-systems.

**Proposition C.5** (Trivial, based on [20])**.** *If $\boldsymbol{\tau} = (\tau_1, \ldots, \tau_{2n})$ is a **T-system** on $\Omega \subseteq \mathbb{R}$, then the induced moment function*

$$\hat{\boldsymbol{\tau}}(\mu) = \left( \int_{\mathbb{R}} \tau_1(x) d\mu(x), \ldots, \int_{\mathbb{R}} \tau_{2n}(x) d\mu(x) \right)$$

*is injective on $\mathcal{M}_{\leq n}(\Omega)$.*

*Proof.* Let $\mu, \mu' \in \mathcal{M}_{\leq n}(\Omega)$ such that $\hat{\boldsymbol{\tau}}(\mu) = \hat{\boldsymbol{\tau}}(\mu')$. Then $\boldsymbol{\tau}(\mu - \mu') = 0$, where $\mu - \mu'$ denotes the difference measure. Since $\mu - \mu'$ is supported on at most $2n$ points, we can write

$$\mu - \mu' = \sum_{i=1}^{2n} w_i \delta_{\boldsymbol{x}_i},$$

where the points $x_i$ are pairwise distinct, and some of the $w_i$ may be zero. Our goal is to show that the vector

$$\boldsymbol{w} = (w_1, \ldots, w_{2n})$$

is all zero. Indeed, the fact that $\hat{\tau}(\mu - \mu') = 0$ implies that $M_{\boldsymbol{\tau}} \cdot \boldsymbol{w} = 0$. And since by assumption $M_{\boldsymbol{\tau}}$ is full rank, we have that $\boldsymbol{w} = 0$. $\qquad\square$

**Example C.6** (Sigmoid T-systems). We've seen that the standard monomial basis forms a T-system. Another well-known example [20] is the family of functions

$$\tau_i(x) = \frac{1}{x - a_i}, \quad i = 1, \ldots, k.$$

If the numbers $a_1, \ldots, a_k$ are pairwise distinct, then $\boldsymbol{\tau} = (\tau_1, \ldots, \tau_k)$ form a T-system on $\Omega = \mathbb{R} \setminus \{a_1, \ldots, a_k\}$.

We can use these example to obtain an alternative T-system based on the sigmoid activation $\sigma(x) = (1 + e^{-x})^{-1}$. Firstly, using the injectivity of the exponent function, we can deduce from the fact that $\boldsymbol{\tau}$ is a T-system, that for any distinct $b_1, \ldots, b_k$ the functions

$$\hat{\tau}_i(x) = \frac{1}{e^{b_i} + e^{-x}}, \quad i = 1, \ldots, k$$

form a T-system on $\Omega = \mathbb{R}$. It follows that the functions

$$\sigma(x + b_i) = \frac{1}{1 + e^{-x - b_i}} = e^{-b_i} \left( \frac{1}{e^{b_i} + e^{-x}} \right), \quad i = 1, \ldots, k$$

form a T-system as well. In particular, for all pairwise-distinct $b_1, \ldots, b_{2n}$, the function

$$x \mapsto \sigma(x + b_i), \quad i = 1, \ldots, 2n$$

is moment injective on $\mathcal{M}_{\leq n}(\mathbb{R})$.

Unfortunately, these results cannot be extended to $\mathcal{M}_{\leq n}(\mathbb{R}^d)$ with $d > 1$, as it is known that T-systems can only be defined on subsets of $\mathbb{R}$ or $S^1$ [20].

# D Proofs

In this section, we provide the proofs omitted from the main text, except for the proofs of Theorem 3.4 and Proposition 3.6, which appear in Appendix A.6.

## D.1 Proofs for Section 3

**Proposition 3.2.** *Let $\sigma : \mathbb{R} \to \mathbb{R}$ be a continuous function that is not a polynomial; then $\sigma$ is discriminatory.*

*Proof.* Let $\sigma$ be as in the statement of the proposition, and let $\mu$ be a signed Borel measure that is finite and is supported on a compact set $K \subseteq \mathbb{R}^d$. Furthermore, assume that $\int_{\mathbb{R}^d} \sigma(\boldsymbol{a} \cdot \boldsymbol{x} + b) d\mu(\boldsymbol{x}) = 0$ for all $\boldsymbol{a}$ and $b$. We need to prove that $\mu = 0$.

Note that by the linearity of $\mu$, we have that $\int f d\mu = 0$ for every $f$ in the space spanned by functions of the form $\sigma(\boldsymbol{a} \cdot \boldsymbol{x} + b)$. Additionally, since functions in this space can approximate any continuous function on $K$ uniformly ([32]), Propositions 3.3 and 3.8), and $\mu$ is compactly supported, we have that $\int f d\mu = 0$ for every continuous function. Finally, by the Riesz representation theorem ([36], Theorem 6.19) we know that a signed (and more generally complex) measure on $K$ is defined uniquely by the integrals of continuous functions and thus $\mu = 0$ as required. $\qquad\square$

**Proposition 3.5.** *Let $n, d \in \mathbb{N}$ and set $m = 2n(d + 1) + 1$. Let $\mathbb{W} = (\boldsymbol{y}, \sigma) \in \mathbb{R}^d \times \mathbb{R}_+$. Then for Lebesgue almost any $(\boldsymbol{y}_i, \sigma_i)_{i=1}^m \in \mathbb{W}^m$, the function*

$$f(\boldsymbol{x}) = \left( \exp\left( -\frac{\|\boldsymbol{x} - \boldsymbol{y}_1\|^2}{\sigma_1^2} \right), \dots, \exp\left( -\frac{\|\boldsymbol{x} - \boldsymbol{y}_m\|^2}{\sigma_m^2} \right) \right)$$

*is moment injective on $\mathcal{M}_{\leq n}(\mathbb{R}^d)$.*

*Proof.* The proof follows the proof of Theorem 3.3 with some modifications. Set $m = 2n(d+1)+1$. A measure in $\mathcal{M}_{\leq n}(\mathbb{R}^d)$ is determined (albeit not uniquely) by a matrix $\boldsymbol{X} = (\boldsymbol{x}_1, \dots, \boldsymbol{x}_n) \in \mathbb{R}^{d \times n}$ that represents $n$ points in $\mathbb{R}^d$, and a weight vector $\boldsymbol{w} \in \mathbb{R}^n$. Let $\mathbb{M}$ denote the space of pairs of measure parameters

$$\mathbb{M} = \{(\boldsymbol{w}, \boldsymbol{w}', \boldsymbol{X}, \boldsymbol{X}') \in \mathbb{R}^n \times \mathbb{R}^n \times \mathbb{R}^{d \times n} \times \mathbb{R}^{d \times n}\},$$

and let

$$F(\boldsymbol{w}, \boldsymbol{w}', \boldsymbol{X}, \boldsymbol{X}'; \boldsymbol{y}, \sigma) = \sum_{i=1}^n w_i \exp\left( -\sigma^{-2} \|\boldsymbol{x}_i - \boldsymbol{y}\|^2 \right) - \sum_{i=1}^n w_i' \exp\left( -\sigma^{-2} \|\boldsymbol{x}_i' - \boldsymbol{y}\|^2 \right). \quad (29)$$

We prove the proposition by showing that for Lebesgue almost every $\boldsymbol{y}_1, \dots, \boldsymbol{y}_m$ and $\sigma_1, \dots, \sigma_m$,

$$\left\{ (\boldsymbol{w}, \boldsymbol{w}', \boldsymbol{X}, \boldsymbol{X}') \in \mathbb{M} \mid \sum_{i=1}^n w_i \delta_{\boldsymbol{x}_i} = \sum_{i=1}^n w_i' \delta_{\boldsymbol{x}_i'} \right\}$$

$$\overset{(*)}{=} \left\{ (\boldsymbol{w}, \boldsymbol{w}', \boldsymbol{X}, \boldsymbol{X}') \in \mathbb{M} \mid F(\boldsymbol{w}, \boldsymbol{w}', \boldsymbol{X}, \boldsymbol{X}'; \boldsymbol{y}, \sigma) = 0, \ \forall \boldsymbol{y} \in \mathbb{R}^d, \sigma \in \mathbb{R}_+ \right\}$$

$$\overset{(**)}{=} \left\{ (\boldsymbol{w}, \boldsymbol{w}', \boldsymbol{X}, \boldsymbol{X}') \in \mathbb{M} \mid F(\boldsymbol{w}, \boldsymbol{w}', \boldsymbol{X}, \boldsymbol{X}'; \boldsymbol{y}_i, \sigma_i) = 0, \ \forall i = 1, \dots, m \right\}$$

As in the proof of Theorem 3.3, equality (**) follows from the *finite witness theorem*. The equality (*) follows on the one hand from the fact that whenever the measures defined by $(\boldsymbol{w}, \boldsymbol{X})$ and $(\boldsymbol{w}', \boldsymbol{X}')$ are the same, necessarily all integrals of functions against these measures are the same, and so $F(\boldsymbol{w}, \boldsymbol{w}', \boldsymbol{X}, \boldsymbol{X}'; \boldsymbol{y}, \sigma) = 0$ for all $\boldsymbol{y} \in \mathbb{R}^d, \sigma \in \mathbb{R}_+$.

On the other hand, if the measure $\mu$ defined by $(\boldsymbol{w}, \boldsymbol{X})$ and the measure $\mu'$ defined by $(\boldsymbol{w}', \boldsymbol{X}')$, are *not* the same, then it is sufficient to show that for some choice of $\boldsymbol{y} \in \mathbb{R}^d, \sigma \in \mathbb{R}_+$ we have $F(\boldsymbol{w}, \boldsymbol{w}', \boldsymbol{X}, \boldsymbol{X}'; \boldsymbol{y}, \sigma) \neq 0$. To see this is indeed the case, choose some $\boldsymbol{y}_0 \in \mathbb{R}^d$ which the non-zero matrix $\mu - \mu'$ assigns a non-zero weight $w_0$. Then

$$\lim_{\sigma \to 0, \sigma > 0} F(\boldsymbol{w}, \boldsymbol{w}', \boldsymbol{X}, \boldsymbol{X}'; \boldsymbol{y}_0, \sigma) = w_0 \neq 0.$$

and so for small enough $\sigma$ we have that $F(\boldsymbol{w}, \boldsymbol{w}', \boldsymbol{X}, \boldsymbol{X}'; \boldsymbol{y}_0, \sigma)$ which concludes the proof. $\qquad\square$

## D.2 Proofs for Section 5

**Proposition 5.1.** *Let $n \geq 2$, $d, m \in \mathbb{N}$, and let $f : \mathbb{R}^d \to \mathbb{R}^m$ be differentiable at some $x_0 \in \mathbb{R}^d$. Then the induced moment function $\hat{f} : \mathcal{S}_{\leq n}(\mathbb{R}^d) \to \mathbb{R}^m$ defined in (1) is not bi-Lipschitz.*

*Proof.* We choose some arbitrary $d \in \mathbb{R}^d$ with unit norm, and focus on multisets of two elements in $\mathbb{R}^d$ of the form $S_\epsilon = \{\!\!\{ x_0 + \epsilon d, x_0 - \epsilon d \}\!\!\}$. We note that the Wasserstein distance $W_2(S_\epsilon, S_0)$ is $\sqrt{2}\epsilon$. Thus, it is sufficient to show that

$$\lim_{\epsilon \to 0} \frac{\|\hat{f}(S_\epsilon) - \hat{f}(S_0)\|}{|\epsilon|} = 0, \tag{30}$$

for this implies that there is no positive $c$ for which (8) holds. Indeed, denoting the differential of $f$ at $x_0$ by $J$, we have

$$
\begin{aligned}
\frac{\|\hat{f}(S_\epsilon) - \hat{f}(S_0)\|}{|\epsilon|} &= \frac{\|f(x_0 + \epsilon d) + f(x_0 - \epsilon d) - 2f(x_0)\|}{|\epsilon|} \\
&= \frac{\|f(x_0 + \epsilon d) - f(x_0) - \epsilon J d + f(x_0 - \epsilon d) - f(x_0) + \epsilon J d\|}{|\epsilon|} \\
&\leq \frac{\|f(x_0 + \epsilon d) - f(x_0) - \epsilon J d\|}{|\epsilon|} + \frac{\|f(x_0 - \epsilon d) - f(x_0) - (-\epsilon J d)\|}{|\epsilon|} \overset{\epsilon \to 0}{\to} 0,
\end{aligned}
$$

where the convergence to zero of the last expression follows from the definition of differentiability at a point. $\qquad\square$

## D.3 Proofs for Section 6

We now prove Corollary 6.1:

**Corollary 6.1.** *Let $n, d \in \mathbb{N}$ and set $m = 2nd + 1$. Let $\sigma : \mathbb{R} \to \mathbb{R}$ be an analytic non-polynomial function. Let $K \subseteq \mathbb{R}^d$ be a compact set. Then there exist $A \in \mathbb{R}^{m \times d}, b \in \mathbb{R}^d$ such that for any continuous permutation-invariant $f : K^n \to \mathbb{R}$, there exists a continuous $F : \mathbb{R}^m \to \mathbb{R}$ such that*

$$f(X) = F\left(\sum_{j=1}^n \sigma(A x_j + b)\right), \quad \forall X = (x_1, \ldots, x_n) \in K^n. \tag{9}$$

*Proof.* By Proposition 1.3 in [9], it is sufficient to show that there exists $A \in \mathbb{R}^{m \times d}, b \in \mathbb{R}^d$ such that the permutation invariant function

$$(x_1, \ldots, x_n) \mapsto \sum_{j=1}^n \sigma(A x_j + b) \tag{31}$$

is orbit separating. This means that any two element in $K^n$ that are not related by a permutation will be separated by the function in (31). Any pair of elements not related by a permutation correspond to two distinct multisets in $\mathcal{S}_{\leq n}(\mathbb{R}^d)$ with exactly $n$ points. By Theorem 3.3, for almost every choice of $A, b$ the function in (31) will be injective on $\mathcal{S}_{\leq n}(\mathbb{R}^d)$, and thus for such choice this function is indeed invariant and orbit separating. $\qquad\square$

We now prove Corollary 6.2:

**Corollary 6.2.** *Let $n, d \in \mathbb{N}$ and set $m = 2n(d + 1) + 1$. Let $\sigma : \mathbb{R} \to \mathbb{R}$ be analytic and non-polynomial. Let $K \subseteq \mathbb{R}^d$ be compact. Then there exist $A \in \mathbb{R}^{m \times d}, b \in \mathbb{R}^m$ such that for any continuous (in the 2-Wasserstein sense) $f : \mathcal{P}_{\leq n}(K) \to \mathbb{R}$, there exists a continuous $F : \mathbb{R}^m \to \mathbb{R}$ such that*

$$f(\mu) = F\left(\int_{x \in K} \sigma(A x + b) d\mu(x)\right), \quad \forall \mu \in \mathcal{P}_{\leq n}(K).$$

*Proof.* Our first step is to show that $\mathcal{P}_{\leq n}(K)$ is compact with respect to the Wasserstein metric. Since $\mathcal{P}_{\leq n}(K)$ is the image of the compact set

$$Q := \{(\boldsymbol{w}, \boldsymbol{X}) \in \mathbb{R}^n \times K^n \mid \sum_{i=1}^{n} w_i = 1 \text{ and } w_j \geq 0, , j = 1, \ldots, n\}$$

under the function

$$(\boldsymbol{w}, \boldsymbol{X}) \mapsto \sum_{i=1}^{n} w_i \delta_{\boldsymbol{x}_i},$$

it is sufficient to show that this function is continuous, as the image of a compact set under a continuous map is compact. Thus, given a sequence of $(\boldsymbol{w}^{(k)}, \boldsymbol{X}^{(k)})$ which converges to some $(\boldsymbol{w}, \boldsymbol{X})$, we need to show that in Wasserstein space, the measures $\mu_k := \sum_{i=1}^{n} w_i^{(k)} \delta_{\boldsymbol{x}_i^{(k)}}$ converge to the measure $\mu := \sum_{i=1}^{n} w_i \delta_{\boldsymbol{x}_i}$. Since the Wasserstein distance metrizes the weak topology on measures ([40], Theorem 6.9), it is sufficient to see that the integral of every continuous $s : K \to \mathbb{R}$ against the sequence of measures converge to the limit measure. Indeed:

$$\int_{K} s(\boldsymbol{x}) d\mu^{(k)}(\boldsymbol{x}) = \sum_{i=1}^{n} w_i^{(k)} s(\boldsymbol{x}_i^{(k)}) \to \sum_{i=1}^{n} w_i s(\boldsymbol{x}_i) = \int_{K} s(\boldsymbol{x}) d\mu(\boldsymbol{x})$$

Thus we have shown that $\mathcal{P}_{\leq n}(K)$ is compact.

Next, by Theorem 3.3, for almost every choice of $\boldsymbol{A}, \boldsymbol{b}$ the function

$$q(\mu) := \int_{\boldsymbol{x} \in K} \sigma(\boldsymbol{A}\boldsymbol{x} + \boldsymbol{b}) d\mu(\boldsymbol{x})$$

is injective on $\mathcal{M}_{\leq n}(\mathbb{R}^d)$, and so in particular on the compact subset $\mathcal{P}_{\leq n}(K)$. As $q$ is also continuous, and a continuous injective function defined on a compact set is a homeomorphism, it follows that $q^{-1} : q(\mathcal{P}_{\leq n}(K)) \to \mathcal{P}_{\leq n}(K)$ is continuous. We then have that $f = (f \circ q^{-1}) \circ q$. We can then use Tietze's extension theorem to extend $f \circ q^{-1}$ from its compact domain to a continuous function $F$ defined on all of $\mathbb{R}^m$, and we then obtain $f = F \circ q$ as required. $\quad\square$

**Theorem 6.3.** *Let $n, d, T \in \mathbb{N}$ and let $\Sigma \subseteq \mathbb{R}^d$ be countable. Let $m \geq 1$ be any integer. Let $\sigma : \mathbb{R} \to \mathbb{R}$ be an analytic non-polynomial function. Then for Lebesgue almost any choice of $\boldsymbol{A}^{(t)}, \boldsymbol{b}^{(t)}$ and $\eta^{(t)}$, the MPNN defined in (10) and (11) assigns different global features to any pair of graphs $G_1, G_2 \in \mathcal{G}_{\leq n}(\Sigma)$ that can be separated by $T$ iterations of 1-WL.*

*Proof.* The message passing iterations discussed in the theorem define new node features $\boldsymbol{h}_v^{(t)}$ from the previous node features $\boldsymbol{h}_v^{(t-1)}$ via

$$\boldsymbol{h}_v^{(t)} = \sum_{u \in \mathcal{N}(v)} \sigma \left( \boldsymbol{A}^{(t)} \left( \eta^{(t)} \boldsymbol{h}_v^{(t-1)} + \boldsymbol{h}_u^{(t-1)} \right) + \boldsymbol{b}^{(t)} \right).$$

This can be rewritten as

$$\boldsymbol{m}_{v,u}^{(t)} = \eta^{(t)} \boldsymbol{h}_v^{(t-1)} + \boldsymbol{h}_u^{(t-1)}$$
$$\boldsymbol{h}_v^{(t)} = f^{(t)} \left( \{\!\{ \boldsymbol{m}_{v,u}^{(t)}, \mid u \in \mathcal{N}(v) \}\!\} \right) = \sum_{u \in \mathcal{N}(v)} \sigma \left( \boldsymbol{A}^{(t)} \boldsymbol{m}_{v,u}^{(t)} + \boldsymbol{b}^{(t)} \right), t = 1. \ldots, T$$

The final 'readout' step creates a global graph features from the node features of the last iteration $T$ via

$$\boldsymbol{h}_G = f^{(T+1)} \left( \{\!\{ \boldsymbol{h}_v^{(T)} \mid v \in V \}\!\} \right) = \sum_{v \in V} \sigma \left( \boldsymbol{A}^{(T+1)} \boldsymbol{h}_v^{(T)} + \boldsymbol{b}^{(T+1)} \right)$$

It is known [44] that every pair of graphs $G, \hat{G} \in \mathcal{G}_{\leq n}(\Sigma)$ which are not be separated by $T$ iterations of the WL test, will not be separated by our message passing procedure, regardless of the choice of the parameters

$$\boldsymbol{\theta} = \left( \boldsymbol{A}^{(1)}, \ldots, \boldsymbol{A}^{(T+1)}, \boldsymbol{b}^{(1)}, \ldots, \boldsymbol{b}^{(T+1)}, \eta^{(1)} \ldots, \eta^{(T)} \right).$$

We need to prove the opposite direction.

Since the collection of all graph-pairs from $\mathcal{G}_{\leq n}(\Sigma)$ is countable, it is sufficient to show that any fixed pair of graphs $G, \hat{G} \in \mathcal{G}_{\leq n}(\Sigma)$ that *can* be separated by $T$ iterations of the WL test, can be separated by *almost every* choice of parameters.

Let us fix such a pair $G, \hat{G} \in \mathcal{G}_{\leq n}(\Sigma)$ which can be separated by $T$ iterations of WL. Note that the final features $h_G = h_G(\boldsymbol{\theta})$ and $h_{\hat{G}} = h_{\hat{G}}(\boldsymbol{\theta})$ obtained from the message passing procedures are an analytic function of the parameters $\boldsymbol{\theta}$, and therefore to show separation for *almost every* $\boldsymbol{\theta}$ it is sufficient to show *existence* of $\boldsymbol{\theta}$ for which $h_G(\boldsymbol{\theta}) \neq h_{\hat{G}}(\boldsymbol{\theta})$.

To show that a single such $\boldsymbol{\theta}$ exists, we choose the parameters of the functions recursively in the order $(\eta^{(1)}, \boldsymbol{A}^{(1)}, \boldsymbol{b}^{(1)}), (\eta^{(2)}, \boldsymbol{A}^{(2)}, \boldsymbol{b}^{(2)}), \ldots$ in which they are applied in the message passing procedure. When choosing the parameters $(\eta^{(t)}, \boldsymbol{A}^{(t)}, \boldsymbol{b}^{(t)})$corresponding to the $t$th step, it is sufficient to show that, if at the previous $(t-1)$ timestamp, for two different nodes $v, w$ we had

$$\boldsymbol{h}_v^{(t-1)} \neq \boldsymbol{h}_w^{(t-1)} \text{ or } \{\!\{\boldsymbol{h}_u^{(t-1)},\ u \in \mathcal{N}(v)\}\!\} \neq \{\!\{\boldsymbol{h}_u^{(t-1)},\quad u \in \mathcal{N}(w)\}\!\} \tag{32}$$

then $\boldsymbol{h}_v^{(t)}$ and $\boldsymbol{h}_w^{(t)}$ will not be the same. Here $v$ and $w$ are nodes in either $G$ or $\hat{G}$.

Our first goal is to choose $\eta^{(t)}$ so that, for all given nodes $v, w$ in $G$ or $\hat{G}$ such that (32) is satisfied, we have

$$\{\!\{\boldsymbol{m}_{v,u}^{(t)},|\ u \in \mathcal{N}(v)\}\!\} \neq \{\!\{\boldsymbol{m}_{w,x}^{(t)},|\ x \in \mathcal{N}(w)\}\!\}. \tag{33}$$

To choose $\eta^{(t)}$ in this way, we note that since all previous parameters were already determined, and since we are only interested in a single pair of graphs, there is only a finite number of features

$$\Sigma^{(t)} = \{\boldsymbol{h}_v^{(t-1)} | v \text{ is a node in } G \text{ or } \hat{G}\}$$

which we are interested in. Since for fixed vectors $\boldsymbol{x}_1, \boldsymbol{x}_2, \boldsymbol{y}_1, \boldsymbol{y}_2$ of the same dimension with $\boldsymbol{x}_1 \neq \boldsymbol{x}_2$, there can be at most a single $\eta$ satisfying the equation

$$\eta\boldsymbol{x}_1 + \boldsymbol{y}_1 = \eta\boldsymbol{x}_2 + \boldsymbol{y}_2. \tag{34}$$

we conclude that all but a finite number of $\eta$ satisfy

$$\eta\boldsymbol{h}_v^{(t-1)} + \boldsymbol{h}_u^{(t-1)} \neq \eta\boldsymbol{h}_w^{(t-1)} + \boldsymbol{h}_x^{(t-1)},$$
$$\forall \boldsymbol{h}_v^{(t-1)}, \boldsymbol{h}_u^{(t-1)}, \boldsymbol{h}_w^{(t-1)}, \boldsymbol{h}_x^{(t-1)} \in \Sigma^{(t-1)} \text{ such that } \boldsymbol{h}_v^{(t-1)} \neq \boldsymbol{h}_w^{(t-1)}$$

We choose $\eta^{(t)}$ which satisfies the inequality above. This implies that if the left-hand side in (32) is indeed an inequality $\boldsymbol{h}_v^{(t-1)} \neq \boldsymbol{h}_w^{(t-1)}$, then $\boldsymbol{m}_{v,u}^{(t)} \neq \boldsymbol{m}_{w,x}^{(t)}$ for all $(u, x) \in \mathcal{N}(v) \times \mathcal{N}(w)$, which in turn implies the inequality of multisets obtained from $w$ and $v$ ((33)). On the other hand, if the left-hand side in (32) is an equality $\boldsymbol{h}_v^{(t-1)} = \boldsymbol{h}_w^{(t-1)}$ and the multisets in the right-hand side of (32) are distinct, then clearly the multisets in (33) are distinct too. Thus, we have proved that we can choose $\eta^{(t)}$ so that (32) implies (33).

It remains to show that we can choose the parameters of $f^{(t)}$ so that, if (33) holds for some nodes $v, w$ of $G$ or $\hat{G}$, then the features $\boldsymbol{h}_v^{(t)}$ and $\boldsymbol{h}_w^{(t)}$ obtained from applying $f^{(t)}$ to the multisets in (33) will be distinct. Indeed, since we are only requiring $f^{(t)}$ to be injective on a finite collection of finite sets, we have that all these multisets are contained in $\mathcal{S}_{\leq n}(\Sigma)$ for some finite $\Sigma$, and therefore by Theorem 3.3 there is a choice of parameters $(\boldsymbol{A}^{(t)}, \boldsymbol{b}^{(t)})$ which is injective on $\mathcal{S}_{\leq n}(\Sigma)$, even when $m = 1$. Thus, we obtained a recursive procedure for choosing the parameters $\boldsymbol{\theta}$ such that $G$ and $\hat{G}$ will be separated, which concludes our proof. $\qquad\square$

# E Numerical Experiments: Additional Details

## E.1 Empirical injectivity and Bi-Lipschitzness

To generate the results shown in Figure 2 (and Figure 4 below), we ran multiple independent test instances in which we generated two random matrices $X_1, X_2 \in \mathbb{R}^{d \times n}$, representing two sets of $n$ vectors in $\mathbb{R}^d$. With exact details appearing below, $X_1, X_2$ were generated such that: (1) Each entry of $X_1$ and $X_2$ has expectation zero and a standard deviation (STD) of 1; (2) $X_1$ and $X_2$ differ in exactly $n_\Delta$ randomly chosen columns, with the parameter $n_\Delta$ chosen uniformly at random from $\{1, \ldots, n\}$; (3) each entry of the $n_\Delta$ nonzero columns of $\Delta X = X_2 - X_1$ has expectation zero and STD=$\rho$, with the parameter $\rho$ itself drawn uniformly (once per test instance) from $[\rho_{\min}, \rho_{\max}]$. We used $\rho_{\min} = 0.02$, $\rho_{\max} = 1$. In other words, the relative difference $\rho$ between non-identical columns of $X_1$ and $X_2$ is chosen at each instance randomly between $2\%$ and $100\%$. The motivation for this construction was to test various types and magnitudes of differences between multisets.

We then randomly generated $\bar{A} \in \mathbb{R}^{\bar{m} \times d}$ and $\bar{b} \in \mathbb{R}^{\bar{m} \times 1}$, from which we took subblocks to be used as parameters for the function $f(x; A, b)$ of Equation (3): for various values of $m \in [\bar{m}]$, we took $A_m, b_m$ to be the top $m$ rows of $\bar{A}$ and $\bar{b}$ respectively, and used them to construct the embedding $\hat{f}(X; A_m, b_m, \sigma) : \mathbb{R}^{d \times n} \to \mathbb{R}$:

$$\hat{f}(X; A_m, b_m, \sigma) = \sum_{i=1}^{n} \sigma(A_m x_i + b_m),$$

with $x_i$ denoting the columns of $X$. The entries of $\bar{A}$ and $\bar{b}$ were drawn from Gaussian distributions chosen such that for each row $a_k$ of $\bar{A}$ and corresponding entry $b_k$ of $b$, and each column $x_i$ of $X_1$ or $X_2$, the input to the activation $\sigma$, $a_k \cdot x_i + b_k$, has expectation zero and STD=1; specifically,

$$\mathbb{E}[a_k \cdot x_i] = \mathbb{E}[b_k] = 0 \quad \text{and} \quad \text{STD}[a_k \cdot x_i] = \text{STD}[b_k] = \tfrac{1}{\sqrt{2}}.$$

For various activation functions $\sigma$, we calculated the ratio

$$r(X_1, X_2) = \frac{\|\hat{f}(X_1; A_m, b_m, \sigma) - \hat{f}(X_2; A_m, b_m, \sigma)\|_2}{W_2(X_1, X_2)},$$

with $W_2(\cdot, \cdot)$ being the 2-Wasserstein distance. We used a Sinkhorn approximation of $W_2(\cdot, \cdot)$, calculated by the `GeomLoss` Python library [11]. Finally, for each activation $\sigma$ and embedding dimension $m$, we took $c$ and $C$ to be the minimum and maximum of $r(X_1, X_2)$ respectively over all test instances, and recorded the ratio $c/C$. In each experimental setting, we ran between 500,000 and 2 million independent instances, depending on the values of $d$ and $n$. The results appear in Figure 4. It can be seen that a similar behaviour is exhibited across different values of $d$ and $n$. In particular, all PwL activations have $c/C = 0$ at low $m$ and all analytic activations have positive $c/C$ in all settings tested.

**Probabilistic distributions of data**   At each instance, as mentioned above, we first chose $n_\Delta \sim$ Uniform $[\{1, \ldots, n\}]$ and $\rho \sim$ Uniform $[\rho_{\min}, \rho_{\max}]$. We then randomly chose $n_\Delta$ columns labelled by $J$, at which $X_1, X_2$ should differ. Let $I = [n] \setminus J$. Denote by $X[:, \Lambda]$ the subblock of the matrix $X$ with columns indexed by $\Lambda$. The entries of $X_1[:, I] = X_2[:, I]$ were drawn i.i.d. from Normal $(0, 1)$. For $X_1[:, J]$ and $X_2[:, J]$, we generated two random matrices $U, V \in \mathbb{R}^{d \times n_\Delta}$, each of whose entries are i.i.d. Gaussian with expectation 0, STD=$\sqrt{1 - \frac{1}{12}(\rho_{\max}^2 + \rho_{\max}\rho_{\min} + \rho_{\min}^2)}$ and STD=1 for $U, V$ respectively. We then set

$$X_1[:, J] = U - \tfrac{1}{2}\rho V, \quad X_2[:, J] = U + \tfrac{1}{2}\rho V.$$

Lastly, we generated the entries of $\bar{A}$ and $\bar{b}$ from Normal $\left(0, \frac{1}{\sqrt{2d}}\right)$ and Normal $\left(0, \frac{1}{\sqrt{2}}\right)$ respectively. We now show that: (i) each entry of $X_1[:, J]$ and $X_2[:, J]$ has expectation zero and STD=1, (ii) given $\rho$, each entry of $\Delta X[:, J]$ has expectation 0 and STD=$\rho$, and (iii) All inputs to the activation $\sigma$ have expectation zero and STD=1. Let $x_1, x_2$ be two corresponding entries of $X_1[:, J], X_2[:, J]$, and let $u, v$ be their corresponding entries of $U, V$. Then

$$x_1 = u - \tfrac{1}{2}\rho v, \quad x_2 = u + \tfrac{1}{2}\rho v.$$

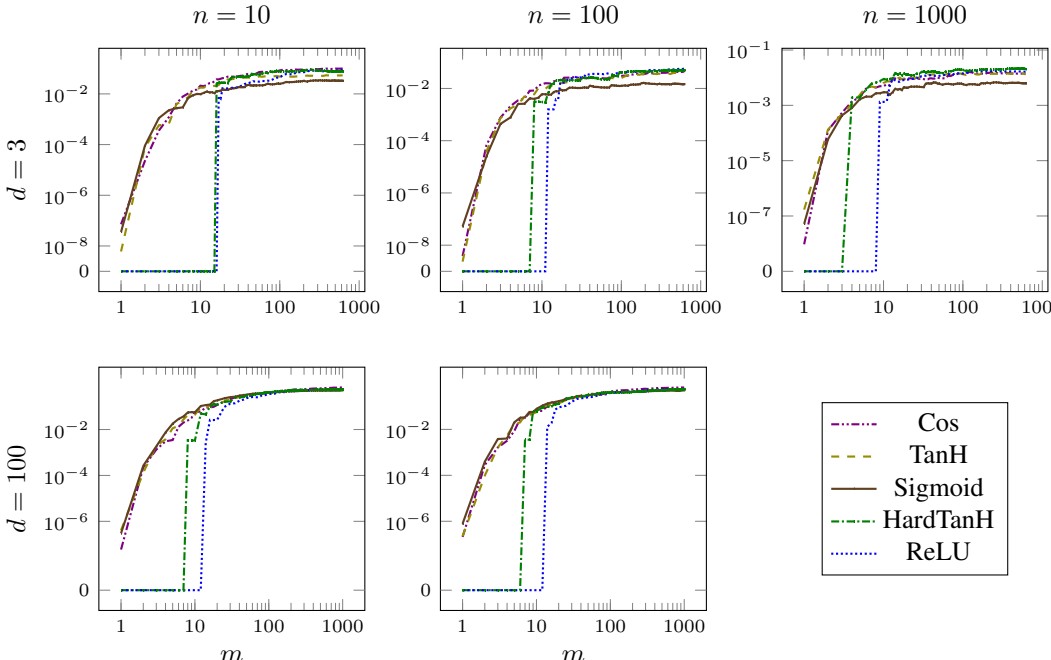

The empirical ratio $c/C$ of Equation (8) as a function of the embedding dimension $m$. The results for different sizes of vector-sets $n$ and ambient dimension $d$ are shown. For low $m$, the piecewise-linear `ReLU` and `HardTanH` have $c/C$ exactly zero. See Appendix E.1 for a full description of the experimental setting.

**Figure 4:** Empirical injectivit and bi-Lipschitzness

We now calculate the expectation and variance of $x_1$, $x_2$. Since $U$, $V$ and $\rho$ are independent, we have that

$$\mathbb{E}[x_1] = \mathbb{E}\left[u - \tfrac{1}{2}\rho v\right] = \mathbb{E}[u] - \tfrac{1}{2}\mathbb{E}[\rho]\mathbb{E}[v] = 0 - \tfrac{1}{2}\frac{\rho_{\min} + \rho_{\max}}{2} \cdot 0 = 0,$$

and by a similar reasoning, $\mathbb{E}[x_2] = 0$. The variance of $x_1$ is given by:

$$\text{Var}[x_1] = \text{Var}\left[u - \tfrac{1}{2}\rho v\right] = \text{Var}[u] + \tfrac{1}{4}\left(\text{Var}[\rho] + \mathbb{E}[\rho]^2\right)\left(\text{Var}[v] + \mathbb{E}[v]^2\right) - \mathbb{E}[\rho]^2\mathbb{E}[v]^2$$

$$= \text{Var}[u] + \tfrac{1}{4}\left(\frac{(\rho_{\max} - \rho_{\min})^2}{12} + \frac{(\rho_{\max} + \rho_{\min})^2}{4}\right)(1 + 0) - \mathbb{E}[\rho]^2 \cdot 0$$

$$= \text{Var}[u] + \tfrac{1}{12}\left(\rho_{\max}^2 + \rho_{\max}\rho_{\min} + \rho_{\min}^2\right)$$

$$= \left(1 - \tfrac{1}{12}\left(\rho_{\max}^2 + \rho_{\max}\rho_{\min} + \rho_{\min}^2\right)\right) + \tfrac{1}{12}\left(\rho_{\max}^2 + \rho_{\max}\rho_{\min} + \rho_{\min}^2\right) = 1.$$

By a similar argument, $\text{Var}[x_2] = 1$. Thus, (i) holds. Let $\Delta x = x_2 - x_1 = \rho v$. Then

$$\mathbb{E}[\Delta x] = \mathbb{E}[x_2] - \mathbb{E}[x_1] = 0 - 0 = 0.$$

Moreover, the conditional variance of $\Delta x$ given $\rho$ is

$$\text{Var}[\Delta x \mid \rho] = \text{Var}[\rho v \mid \rho] = \rho^2 \cdot \text{Var}[v \mid \rho] = \rho^2 \cdot \text{Var}[v] = \rho^2,$$

and thus (ii) holds. Finally, we show that (iii) holds. We already have established that each entry of $X_1$ and of $X_2$ has zero mean and STD=1. Let $a_k$, $b_k$ be a row or $\bar{A}$ and its corresponding entry of $\bar{b}$. Let $x_i$ be an arbitrary column of $X_1$ or $X_2$. Then

$$\mathbb{E}[a_k \cdot x_i] = \mathbb{E}\left[\sum_{j=1}^{d}(a_k)_j \cdot (x_i)_j\right] = \sum_{j=1}^{d}\mathbb{E}\left[(a_k)_j\right]\mathbb{E}\left[(x_i)_j\right] = \sum_{j=1}^{d} 0 \cdot 0 = 0,$$

and by definition $\mathbb{E}\left[b_k\right] = 0$. Since each $(a_k)_j$ and $(x_i)_j$ are independent random variables with expectation zero, we have that

$$\mathrm{Var}\left[a_k \cdot x_i\right] = \mathrm{Var}\left[\sum_{j=1}^{d}(a_k)_j \cdot (x_i)_j\right] = \sum_{j=1}^{d}\mathrm{Var}\left[(a_k)_j\right]\mathrm{Var}\left[(x_i)_j\right] = \sum_{j=1}^{d}\tfrac{1}{2d}\cdot 1 = \tfrac{1}{2},$$

and by definition $\mathrm{Var}\left[b_k\right] = \tfrac{1}{2}$. Therefore, (iii) holds.

**Computational resources**   All experiments were run on an NVidia A40 GPU with 48 GB of GPU memory.

### E.2   Graph separation

We ran the graph separation experiment using the PyTorch Geometric [10] implementation for 'WL convolutions', and the GCN convolutions of [18], as well as their version of the TUDataset [30]. As initialization for the node features, we only took the node degree.

When moving from the abstract mathematical world to finite precision computing, features that are mathematically equal $h_i = h_j$ could end up having slightly different values. To deal with this, all computations were done in double precision. The final features computed by the MPNN were normalized to have an average norm of one, and two features $h_i, h_j$ were deemed equal if $|h_i - h_j| < 10^{-12}$. In Figure 5 we show the minimum and median of the quantity $\|h_i - h_j\|$ over all $(i, j)$ from all graphs for which this norm was larger than the threshold of $10^{-12}$. This is shown for the SiLU activation, and the values are shown as a function of the hidden dimension used. We see that the minimal non-zero distance was safely larger than the threshold in all examples. Also note that the minimal distance moderately improves as the hidden dimension increases.

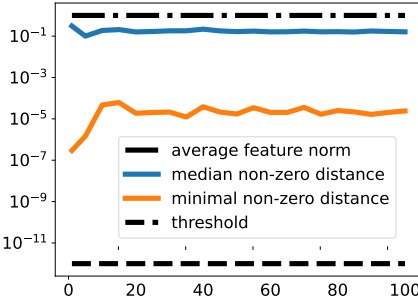

**Figure 5:** Minimal and median distance between non-equivalent features as a function of hidden feature dimension.

