# OpenReview forum: "Neural Injective Functions for Multisets, Measures and Graphs via a Finite Witness Theorem"
_NeurIPS.cc/2023/Conference — NeurIPS 2023 spotlight_

### Official Review · Reviewer_gUtg · 2023-07-06

**Soundness:** 3 good
**Presentation:** 4 excellent
**Contribution:** 3 good
**Rating:** 7
**Confidence:** 3

**Summary:**

This paper studies moment-injective functions defined by neural networks. A moment injective function f can be used to define injective multiset function $g(\{\{x_1, x_2, ..., x_n\}\}) = \sum_{i = 1}^n f(x_i)$. Study of infectivity of multiset functions is motivated by recent developments in Graph neuran networks and Message-Passing Neural networks.

Prior work shows that when $x_i \in \mathbb{R}^d$ moment injective function f should be an embedding of dimension at least nd, moreover one can construct moment-injective polynomial embedding of dimension 2nd+1.

This paper extends prior work by showing that depth 1 neural networks of width at least 2nd+1 with analytic non-polynomial activation unit defines a moment injective embedding for almost all choices of weights. Moreover, the paper shows that the result can be generalized to neural networks of higher depth (from theoretical perspective depth-1 case is the most interesting, as it involves the least amount of parameters).

**Strengths:**

The paper provides theoretical explanation why neural networks are successful in providing injective multiset functions that gained popularity since the seminal Deep-Sets paper. While neural networks are long-known to be universal approximations for $f: R^k -> R^n$. Much less is known about multiset functions represented by neural networks. The main result proven in this paper (Theorem 3.3) shows that depth-1 neural networks can be used to define injective multiset functions, moreover, they achieve currently best-known embedding dimension. The paper also shows that one cannot use piecewise-linear activations to construct injective multiset functions and should use analytic non-polynomial activations instead (for example, sigmoid).

The paper is well-structured and provides a clear comparison to the prior work. The proofs are rigorous and well-written and as far as I can tell are correct.

**Weaknesses:**

I think the paper may benefit from slightly more detailed discussion of applications of injective multiset functions. While injectivity is a natural property that proves to be useful in various setups, I believe it will be nice to include a slightly more detailed discussion of the applications in the introduction. Some example applications are presented in Section 6.

**Questions:**

1) The paper shows that MLPs define injective multiset functions if activation is analytic non-polynomial, and fail to define injective multiset functions if activation is piecewise linear. What happens if activation is polynomial? Is it easy to see that such activations fail similarly to piecewise linear activations?

2) I think it worth replacing "up to a multiplicative factor of 2" everywhere in the paper with "up to a multiplicative factor of 2+o(1)" to be mathematically precise. If one is pedantic, 2nd+1 is not within a factor of 2 from nd.

I believe the paper contains several typos:
L47: a Euclidean -> an Euclidean
L129: this paper -> that paper (?)
L456 quality -> qualify.

---

> ### Author Rebuttal · Authors · 2023-08-08
>
> We thank the reviewer for his/her helpful comments. Below are our responses.
>
> **Response to Weaknesses**
>
> We thank the reviewr for this comment. Should the paper be accepted, we intend to add to the introduction a broader explanation of the importance of injective multiset functions for practical applications.
>
> **Response to Questions**
>
> 1. A possible direction to show how polynomial activations fail would be as follows: If $\sigma : \mathbb{R} \to \mathbb{R}$ is a polynomial of degree $r$, then $\text{span} \\{ \sigma(ax+b) \\, \mid \\, a \in \mathbb{R}^d, b \in \mathbb{R} \\}$ is a subset of the span of all polynomials of degree up to $r$ over $\mathbb{R}^d$. Denote the latter by $\mathcal{P}\_{r}$. Using the fact that $\mathcal{P}\_{r}$ is not dense in $\bigcup\_{t=0}\^{\infty} \mathcal{P}\_{t}$, show that for a large enough $n$ there exist $n$ points $x_1,\ldots,x_n \in \mathbb{R}^d$ and a polynomial $q \in \mathcal{P}\_{r'}$ for some $r'>r$ such that: (a) $\sum_{i=1}^n q(x_i) p(x_i) = 0$ for any $p \in \mathcal{P}\_{r}$; (b) not all of $q(x_1),\ldots,q(x_n)$ are zero. Let $\mu = \sum_{i=1}^n w_i \delta_{x_i}$ be the discrete signed measure with weights $w_i = q(x_i)$. Then (a) implies that for any embedding $\hat{f}$ comprised of moments of $\sigma(ax+b)$, $\hat{f}(\mu) = 0$, although by (b), $\mu$ is not the zero measure, hence injectivity is violated.
>
> 2. We thank the reviewer for this comment. We will replace this statement by a more accurate one.

---

> > ### Comment · Reviewer_gUtg · 2023-08-16
> >
> > I appreciate the authors' response to my questions and keep my positive score unchanged.

---

### Official Review · Reviewer_ejpb · 2023-07-06

**Soundness:** 4 excellent
**Presentation:** 4 excellent
**Contribution:** 3 good
**Rating:** 8
**Confidence:** 4

**Summary:**

In this paper, the authors study the injectivity of functions of multisets, which has garnered a lot of attention recently in the machine learning community following work on point clouds and graphs. Unlike earlier work, which prove results for generic continuous functions or polynomials, then may resort to the universality of MLPs but with unbounded width, here the authors directly focus on MLPs with finite width and prove injectivity with near-optimal width, for almost all parameters. This holds with analytic non-linearity, and on the contrary negative results are given for piecewise linear functions like ReLU. Their result relies mostly on a new "finite witness" theorem, which extend previous results known for semi-algebraic sets and functions to sub-analytic sets and functions. Several corollaries are presented, as well as some illustrative numerical experiments.

**Strengths:**

This is honestly a fantastic paper, I very much enjoy the read, it is clear, pedagogical, and refreshingly honest. The results presented are of great interest for the community. Although it is an extension of a known results on semi-algebraic sets, as acknowledged by the authors, the extension seems anything but trivial. The large part dedicated to negative results and/or limitations is very much appreciated.

**Weaknesses:**

Minor weaknesses; maybe a tad more explanation on the difference between semi-algebraic and sub-analytic proof (I understand both lies on the same underlying proof technique initiated with works on phase retrieval, but with different tools, how would you describe the main difference if possible?), and a bit more outlooks: do you have any more insight what would be a desirable property to show on ReLU networks, since they are definitely not injective, but are used in practice?

**Questions:**

Some questions/minor typos:
- l45: nessecary
- equation (***): missing indices $i$ on the $x$'s
- same equations: might be clearer if the $a$ and $b$ were indexed by $i$ and the $x$'s by $j$ (or vice versa)
- thm 3.4: co-existence of $D$ and $D_\theta$
- eq (6) and lines below: in your definition of Wasserstein, do you mean "uniform" weights instead of "unit"? $S_1$ and $S_2$ may have different cardinality, but the associated measure must be normalized to compute (the classical) Wasserstein
- l 288: seleceted
- l 298: Lipshcitz
- the experiment on bi-Lipschitz is not really illustrative of the theory, since functions are not bi-Lipschitz. It is not uninteresting in describing another phenomenon (the "almost" bi-Lipschitzness on finite data, etc), but I wouldn't say that it "corroborates Thm 5.1"

**Limitations:**

Authors have focused extensively on negative results and limitations.

---

> ### Author Rebuttal · Authors · 2023-08-08
>
> We thank the reviewer for the supportive and helpful comments. Below are our responses to the questions and weaknesses.
>
> **Response to Weaknesses**
>
> 1. We thank the reviewer for the suggestion. We intend to add to the text an explanation of the main challenges and ideas in this generalization.
>
>     In a nutshell, The proof of the finite witness theorem in [Dym and Gortler] relies on several nice properties of semialgebraic sets: This family is closed to linear projections, finite unions, finite intersections, and complements. Moreover, they are always a finite union of smooth manifolds. Using these properties, [Dym and Gortler] prove a finite witness theorem for semialgebraic sets and corresponding functions, called semialgebraic functions. These functions include polynomials, but do not include other analytic functions of interest.
>
>     The generalization of the finite witness theorem to the analytic setting relies on the mathematical study of o-minimal systems, which searches for larger families of sets that enjoy the same properties as semialgebraic sets. There are several known o-minimal systems, such as globally subanalytic sets, which do have the same nice properties. For these sets, the generalization of the finite witness theorem is rather straightforward. However the corresponding collection of globally subanalytic functions still does not include all analytic functions. A crucial observation in the proof is that we can make it carry through also when considering countable unions of globally subanalytic sets. Consequently, the corresponding functions we can work with include all analytic functions.
>
> 2. Although any finite-size ReLU-activated network is not moment injective, it can be shown that moments of shallow ReLU networks are universal approximators of continuous injective functions on multisets; this is due to the ReLU function being discriminatory, as commented below the statement of Corollary 6.1. Thus, taking a high enough embedding dimension, it is possible to construct invariant embeddings using ReLU activations that are practically injective. This comes with a caveat, though, as we noted in our response to Reviewer ezv6: Each such embedding has a nonzero-measure subset of the input domain on which it is provably not injective. Should the paper be accepted, we intend to comment on this in the camera-ready version.
>
>
> **Response to Questions**
>
> We thank the reviewer for the corrections.
>
> 1. In Eq. (6) and below, the term $\text{\emph{uniform}}$ is indeed more suitable, as the proof works well with a total mass of 1.
>
> 2. We rephrased the text describing the bi-Lipschitzness experiment (Line 296) and removed the statement that the results corroborate Thm 5.1.

---

> > ### Comment · Reviewer_ejpb · 2023-08-16
> > **Response to rebuttal**
> >
> > Thank you for the rebuttal. I keep my good score as is.

---

### Official Review · Reviewer_ezv6 · 2023-07-09

**Soundness:** 3 good
**Presentation:** 3 good
**Contribution:** 3 good
**Rating:** 7
**Confidence:** 3

**Summary:**

Injective functions on multisets are commonly employed in the literature for universality results on (multi)set architectures and graph neural network separation results. Usually, the assumption is that MLPs can implement moment injective functions. The paper studies whether MLP architectures are moment-injective in the space of multisets and the conditions that are required to ensure that.

The paper argues that for a (shallow) MLP to be moment injective, its activations have to be analytic and discriminatory. The paper considers a more generalized treatment that views multisets from the perspective of signed discrete measures. The goal of the proof for moment injectivity is to show that a shallow MLP separates all pairs of distinct measures in the space of measure parameters. There are two key conditions for the main theorem :

- the activation of the MLP needs to be analytic and discriminatory, which are classic conditions for MLP results
- the finite witness theorem, which is used to reduce an infinite number of equalities down to a finite one.

The finite witness theorem (and its generalized version) can be leveraged to show the moment-injectivity of various functions.

The paper then examines cases (alphabets) where moment injectivity doesn't hold for piece-wise linear activations. The stability properties (stability is studied through the lens of bi-lipschitzness) of injective multiset functions induced by moment injective MLPs are then studied and it is shown that as long as any moment injective function is differentiable at some point, then the induced multiset function won't be stable (bi-lipschitz).

Finally, the paper provides a couple of applications of the technical results to function approximation and graph separation and provides some experimental evidence to back some of the technical claims up.

**Strengths:**

- The paper contains several interesting mathematical results which could be useful to the broader graph/set NN community.
- The paper is well written and generally provides enough context and guidance for the results to be understandable.
- The theoretical results are focused on architectures that are used in practice.
- The experiments provide additional context and intuition for the applicability of the results.
- The paper is upfront about its scope, limitations, and practical applicability.

**Weaknesses:**

- The initial motivation of the paper is the moment injectivity of 'practical' MLPs. While several results are ultimately established, we can see from the experiment on graph separation (and it is clearly stated in the conclusion) that even non-analytic functions just barely fail in a few cases to match 1-WL. This seems to suggest that the practical relevance of the results is a bit questionable, or maybe there are other factors that could be mitigating the non-analyticity of the activations.

- A few extra explanatory lines could be included in the section of the finite witness theorem (or perhaps in the introduction), that provide context about its exact role in the proof. Currently, the theorem shows up a bit abruptly when theorem 3.3 is presented. OK, it can be inferred from the proof of 3.3 that infinitely many equalities are reduced down to a finite number, but I believe the readability of that section could improve if some of the context that is provided in the supplementary material was moved in the main text to make sure that the role of the theorem is very clearly explained and emphasized. Specifically, when explaining proof 3.3, around line 154 or perhaps right below the statement of 3.4, in subsection 3.1. Those could be good places to provide a few more sentences that restate the purpose of the theorem and explain the context around it (papers by Balan et al., and Dym and Gortler).

I am skeptical about the significance of the results when it comes to practical considerations but the paper is well written and provides mathematical insights that can be relevant to modern deep learning architectures so I lean towards accepting.

**Questions:**

We see in the second experiment a somewhat significant difference between leaky ReLU and ReLU. Could that be a matter of randomness from different initialization, or could the properties of the function that could lead to worse outcomes when it comes to matching the 1-WL? I don't see any obvious reason why the leaky activation would be much different.

---

> ### Author Rebuttal · Authors · 2023-08-08
>
> We thank the reviewer for his/her constructive review. Below we address the weaknesses and questions raised by the reviewer.
>
> **Response to Weaknesses**
>
> 1. While indeed in our experiments ReLU-based embeddings performed similarly to analytical activations when the embedding dimension was high enough, this can be contrasted by the following fact: For any moment function $\hat{f} : \mathcal{S}_{\leq n} (\mathbb{R}^d)$ based on ReLU-activated networks, there exists some $\delta > 0$ and neighbourhood of radius $\delta$ on which $\hat{f}$ is not injective. In contrast, for any $\delta>0$, any injective analytical embedding $\hat{f}$ is guaranteed to be bi-Lipschitz on all pairs of point-sets whose distance is at least $\delta$. Namely, assuming that the domain is compact, for any $\delta>0$ there exist constants $c(\delta),C(\delta) > 0$ such that if $W_2(X_1, X_2) \leq \delta$, then $c(\delta) W_2( X_1, X_2 ) \leq \lVert \hat{f}(X_1) - \hat{f}(X_2) \rVert \leq C(\delta) W_2( X_1, X_2 )$.
>
>     The reason why we did not enounter pathological $\delta$-neighborhoods for ReLU in our experiments with high $m$ is because we drew the input clouds randomly rather than explicitly looking for adversarial examples. We intend to clarify on this in the camera-ready version if our paper is accepted.
>
> 2. We revised Section 3 to clarify the role of the finite witneess theorem in the proof of moment injectivity.
>   In the revised version, the theorem is gently introduced to the reader before proving Theorem 3.3, and only then it is applied to prove the result.
>   Our new proof of Theorem 3.3 is, in our opinion clearer, and places more emphasis on the essential role of the finite witness theorem.
>
> **Response to Questions**
>
> We conjecture that the differences in favor of Leaky ReLU over ReLU (Figure 1) result from the latter having a region where it is identically zero.
>
> An interesting perspective on this is to compare a ReLU, a leaky ReLU and a nonpolynomial analytic activation, by regarding their restrictions to the region x < 0 as polynomials of degree 0, 1 and $\infty$ respectively. In light of Theorem 3.3, it seems plausible that as the degree of the polynomial increases, the likelihood of getting the same output $\hat{f}(X_1)=\hat{f}(X_2)$ for a distinct pair of inputs $X_1,X_2$ should decrease.

---

> > ### Comment · Reviewer_ezv6 · 2023-08-16
> > **Update**
> >
> > Thanks for the rebuttal and the interesting remarks! As far as I am concerned the paper is solid and should be accepted so I maintain my score.

---

### Decision · Program_Chairs · 2023-09-21

**Decision:**

Accept (spotlight)

**Comment:**

This paper provides analytic conditions for an MLP to be moment injective. The reviewers unanimously agreed that this is a great paper and worthy of being accepted to NeurIPS, and I concur.